# Observation of mechanical kink control and generation via acoustic waves

Kai Qian [1,2], Nan Cheng [3,7], Francesco Serafin[3,4,7], Nicolas Herard[1], Kai Sun [3], Georgios Theocharis [5] ✉, Xiaoming Mao [3] ✉ & Nicholas Boechler [1,6] ✉

Kinks are localized transitions between topologically distinct ground states and play a central role in systems from condensed matter to cosmology. While acoustic wave packets (here defined as small-amplitude mechanical waves, sometimes referred to as phonons) have been predicted to drive kink motion deterministically, experimental evidence has been elusive, with only stochastic motion from thermal phonons or quasi-static loading observed. This is largely due to the discrete nature of real materials, where the Peierls-Nabarro (PN) barrier hinders controlled phonon-kink interactions. Here, we report experimental observation of acoustic-wave–mediated control and generation of mechanical kinks in a topological metamaterial, which eliminates the PN barrier by supporting a zero-energy kink. We also computationally reveal the dynamics of acoustic wave packet interplay with highly discrete kinks, including long-duration motion and a continuous family of internal modes—features absent in conventional discrete nonlinear systems. Our results enable remote kink control, with potential applications in material stiffness tuning, shape morphing, locomotion, and robust signal transmission.

Kinks (or domain walls) are localized transitions between distinct ground states that are associated with a topological invariant, and are of great significance in physics, across fields ranging from condensed matter to cosmology[1,2]. While definitions vary depending on the field of research[3], given the one-dimensional (1D) setting considered herein, we will use kinks and domain walls interchangeably. A subset of kinks is what we define as mechanical kinks, wherein the transition between ground states involves a gradient of displacement (*i.e.*, strain). Several settings that see mechanical kinks, spanning a wide range of length and time scales, are crystal plasticity[4,5], formation and propagation of denaturation bubbles in DNA[6], interfaces in polyacetylene[7], ferroelectric domains[8], colloidal self-assembly[9], as well as bistable and topological states in mechanical metamaterials[10–18]. Given their ubiquity and impact on mechanical and transport properties of materials,

understanding how mechanical kinks can be controlled and generated is a topic of significant interest.

Many studies have experimentally demonstrated that acoustic wave packets (here defined as small-amplitude, near-linear mechanical waves, sometimes referred to as phonons in their classical sense[19]) can initiate and affect the movement of mechanical kinks. However, in each of these studies, the waves are either so far below the characteristic frequencies of the system that it can be considered a quasi-static loading[20], or applied in such a spatially extended manner that the resulting kink motion is stochastic, *e.g.*, acoustic annealing in colloidal crystals[9], or thermally-induced motion of dislocations in metals[21] and kinks in polyacetylene[22]. Another, albeit less, related area is acoustic wave interplay with multistable systems supporting transition waves, *e.g.*, ref. 23. We note that these systems do not support kinks as they do

[1]Department of Mechanical and Aerospace Engineering, University of California San Diego, La Jolla, CA, USA. [2]George W. Woodruff School of Mechanical Engineering, Georgia Institute of Technology, Atlanta, GA, USA. [3]Department of Physics, University of Michigan, Ann Arbor, MI, USA. [4]Department of Physics and Materials Science, University of Luxembourg, Esch-sur-Alzette, Luxembourg. [5]Laboratoire d'Acoustique de l'Université du Mans (LAUM), UMR 6613, Institut d'Acoustique - Graduate School (IA-GS), CNRS, Le Mans, France. [6]Program in Materials Science and Engineering, University of California San Diego, La Jolla, CA, USA. [7]These authors contributed equally: Nan Cheng, Francesco Serafin. ✉e-mail: georgios.theocharis@univ-lemans.fr; maox@umich.edu; nboechler@ucsd.edu

not have symmetric ground states and use high-amplitude excitation to initiate transitions. As a result, we ask: Is it possible to control and generate mechanical kinks with acoustic waves in a deterministic manner?

Theoretical and computational studies have given further insight to this question, exploring the interaction between acoustic waves and mechanical kinks via canonical models such as the $\phi^4$[24–31] and sine−Gordon (sG) systems[32]. One challenge in precisely controlling kinks in this manner, as revealed by studies of such models, stems from the fact that all the aforementioned physical systems supporting kinks are, at some level, discrete, where the width of the kinks is comparable with the lattice spacing[4–18]. This is important because of the emergence of the static Peierls−Nabarro (PN) potential barrier in discrete settings[33–38], which arises from the breaking of the translational invariance present in continuum systems[1,37,38]. The PN barrier restricts the mobility of the kinks−typically causing moving kinks to lose energy through phonon radiation and eventually become pinned[39]. Although studies have identified discrete kink models absent of a PN barrier (i.e., barrier-free) arising from exceptional discretizations[38,40–43], these cases are primarily of mathematical interest, with no known physical realization. Beyond controlling mechanical kinks, their generation is also difficult, as it requires initiation either through large-amplitude excitations[44] or within active systems[45–47].

In this work, we report the first experimental observation of mechanical kink control and generation via acoustic wave packets. In particular, the kinks we control and generate are highly discrete (width less than two lattice spacings), which is important, as the strength of the PN barrier typically increases as the kink width decreases[36,48,49]. To achieve this, we created an elastically coupled realization of the Kane−Lubensky (KL) chain model[50]−a 1D topological metamaterial−as shown in Fig. 1. In contrast to the other aforementioned kink-supporting systems, mechanical metamaterials further facilitate experimental observation of such control, as they are typically macroscopic with significantly lower characteristic frequencies[10,13,17,18]. The special feature of the KL chain, as regards kink control via acoustic wave packets, is that it supports a single topologically protected kink that requires zero energy to move quasi-statically (also known as a nonlinear zero mode or a mechanism), resulting in a zero PN barrier[10–12,14,15]. Herein, we refer to this kink as a zero-energy kink.

In addition to experimental observations, we provide a comprehensive numerical study of the discrete modal spectrum of the KL chain featuring the zero-energy kink. We find the existence of several internal modes[1,37,38,51], especially for highly discrete (narrow) kinks. Internal modes have been shown to play a crucial role in kink dynamics, as they can store and release energy, leading to resonance

effects during kink collisions[52] or interactions with inhomogeneities[53]. Experimental evidence of internal modes has been reported in a variety of systems, from polyacetylene[54] to crystals of trapped ions[55], underscoring their broad relevance in different physical contexts. Significantly, in our KL chains, we identify a continuously varying set of finite-frequency internal modes associated with smoothly varying kink states (i.e., kink-containing chain configurations). This continuity is distinct from other nonlinear discrete systems, which typically support only a finite set of kink states with corresponding internal modes, such as those associated with onsite- or intersite-centered kink states[1,37,38].

Besides the discrete modal spectra of the zero-energy kink, we also numerically reveal two other distinctive features of acoustic-wave−kink interaction in the KL chain. First, we observe long-duration kink motion with no apparent slowdown after interaction with an acoustic wave packet. This contrasts with the behavior seen in the discrete $\phi^4$ chain, where the kink eventually slows down and stops due to radiation induced by the PN barrier. The long-lasting motion observed here has important implications for kink control, as it suggests that the kinks could be driven further with less energy. Second, we observe a range of acoustic-wave−kink interactions depending on the geometric properties of the chain, including both kink attraction and repulsion. We also show how these interactions are influenced by factors such as the amplitude and frequency of the wave packet and the kink's center position, thereby providing additional knobs for kink control.

Building upon this capacity for kink control, we highlight here additional application-relevant benefits, as well as potential future physical manifestations. Because the KL chain kink is a topologically protected defect, one can imagine future uses in robust signal transmission[56]. Similarly, because the KL chain kink represents a transition between states of, ideally, localized zero and infinite stiffness[12,57], one can imagine remote control of extreme material stiffness, including gradient properties. Further functionalities enabled by kink motion include locomotion[17] and shape-shifting materials[58], both of which would benefit from low-energy control and long-distance kink transport. Such phenomena may also inspire the search for analogous nanoscale or molecular-level mechanisms[59]. Potential analogies may be drawn with, e.g., rotor-like behaviors in bacterial flagellar motors[60], DNA[61], or nanoelectromechanical systems[62].

## Results
### Model
We start by reviewing the key features of the KL chain and its kink state. We describe the rotor angles with $\psi_n$ (where $n$ is the rotor index), measured clockwise for even rotors and counterclockwise for odd, with the rotor radius, lattice constant, spring stiffness, point mass, and instantaneous length denoted by $r$, $a$, $k_e$, $m$, and $l_{n,n+1}$, respectively, as shown in Fig. 1a. The KL chain exhibits two homogeneous configurations when $\psi_n = \pm\bar{\psi}$ ($\bar{\psi} \in [0, \pi/2]$), referred to as the right- (RPS) and left-polarized state (LPS), respectively, resulting in a uniform unstretched spring length $l_{n,n+1} = \bar{l}$ throughout the chain[10,12,50]. In the linear regime, these configurations give rise to a topological polarization vector $R_T$, leading to an exponentially localized zero-energy mode at the edge pointed to by $R_T$[10,12,50]. This localized zero mode creates a soft edge, with the rest of the chain rigid.

When the zero mode in the homogeneous chain is quasi-statically driven to the nonlinear regime, it smoothly deforms into a kink state that separates regions of opposite polarizations and exhibits zero energy[10], i.e., the kink is soft, as marked in Fig. 1b. This kink exhibits two distinct continuum limits−one described by a field theory resembling $\phi^4$[1,10,38,63] and the other by the sG field theory[1,10,37]−depending on the unit cell geometry[10]. We denote the KL chain kink width as $w_0 = 2\kappa$, where $\kappa = -a/ln|(d-1)/(d+1)|$ is the penetration depth of the zero mode in the homogeneous chain, with $d = 2r\sin\bar{\psi}/a$ a dimensionless geometric index[10]. When $d \ll 1$, $\kappa \approx a/(2d)$ (diverging as $d \to 0$), and the kink can be described by the continuum theory similar to $\phi^4$[10]. When

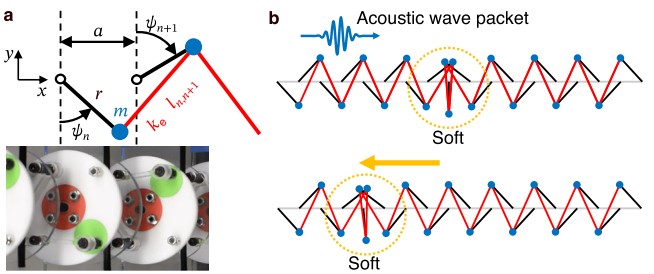

**Fig. 1 | Overview of a small-amplitude acoustic wave packet moving the static zero-energy kink in the KL chain. a** Schematic of a chain section, where black lines represent massless rotors with radius $r$ and lattice constant $a$, blue circles represent point masses $m$, and red lines represent linear normal springs with spring constant $k_e$ and instantaneous length $l_{n,n+1}$ (with $n$ and $n+1$ as rotor indices), accompanied by a photograph of the corresponding experimental chain section. **b** Kink state before (top) and after (bottom) acoustic-wave−kink interaction. The yellow dashed circle indicates the center of the kink.

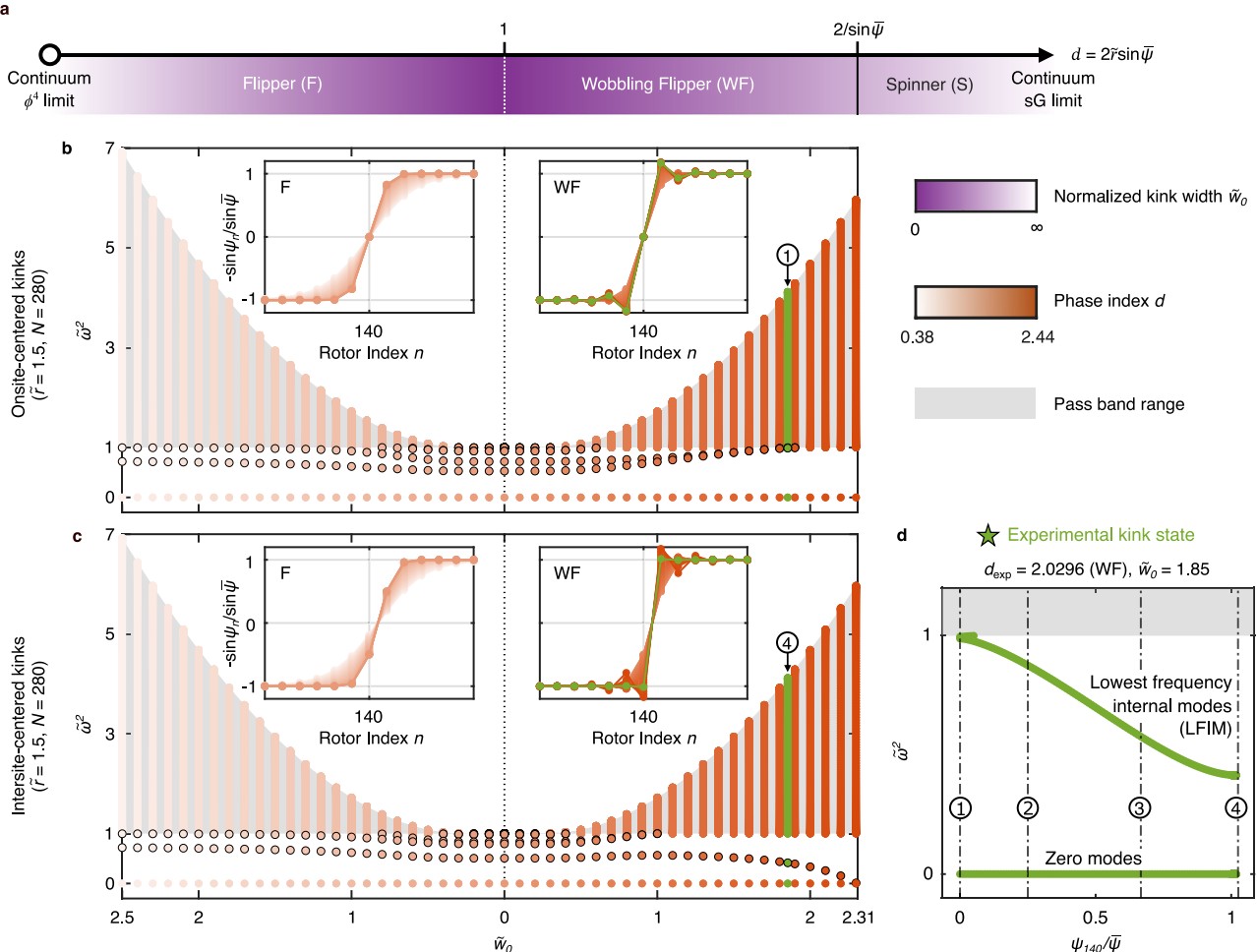

**Fig. 2 | Computed discrete modal spectra of KL chain's F/WF-phase kinks, including a family of internal modes with smooth variation between sites.** **a** Phase and kink width diagram of the zero-energy kink in the KL chain as a function of the dimensionless index $d$[10]. **b**, **c** Normalized eigenfrequencies of kink states as a function of their normalized widths $\tilde{w}_0$ for KL chains with $\tilde{r} = 1.5$ and $N = 280$ rotors, where (**b**) and (**c**) show results for onsite- ($\psi_{140} = 0$) and intersite-centered ($\psi_{140} = -\psi_{141}$) kinks, respectively. Markers with black edge indicate internal modes. Insets show zoomed-in, normalized kink states ($-\sin\psi_n/\sin\bar{\psi}$[10]).

Green color denotes the kink state examined in experiments, which is further used in (**d**), as well as Figs. 3 and 4, to denote the same geometry. For F- and WF-phase kinks, the lowest eigenfrequency remains zero, indicating barrier-free propagation independent of kink width (*i.e.*, discretization). **d** In-gap modes of the experimental kink state as a function of the kink's center rotor angle. Circled marker 1 and 4 indicate onsite- and intersite-centered kinks, respectively, while circled marker 2 and 3 correspond to intermediate kink states between them, which are $\psi_{140}/\bar{\psi} = 1/4$ and 2/3 (further used in Figs. 3 and 4).

$d \gg 1$, $\kappa \approx ad/2$ (diverging as $d \to \infty$), and the kink can be described by the continuum sG theory, where the representation involves only alternating rotors[10]. Possessing two continuum limits is distinct from kinks in other systems, such as a discrete $\phi^4$ chain[1], which exhibits a single continuum limit for wavelengths much longer than unit cell size.

Between these two continuum limits, the kink state can be classified into three distinct phases[10] based on the range of rotor angle changes during kink propagation, characterized by $d$: i) the Flipper (F), where $d < 1$ and $\psi_n$ varies between $-\bar{\psi}$ and $+\bar{\psi}$; ii) the Wobbling Flipper (WF), where $1 < d < 2/\sin\bar{\psi}$ and the rotors move beyond their homogeneous positions but cannot complete full rotations; and iii) the Spinner (S), where the rotors can complete full rotations. We note that the continuum limits occur deep within the F and S phases, as illustrated in Fig. 2a. In Fig. 2a, we also illustrate the dependence of normalized kink widths $\tilde{w}_0 \equiv w_0/a$ on $d$ and highlight the difference between the F/WF- and S-phase kinks by a solid line, which indicates the topological change in the configuration space of the KL chain in the linkage limit[10]. Herein, we investigate F- and WF-phase kinks, and leave the investigation of S-phase kinks for future work. Noting that the unit cell geometry for a given phase $d$ is not unique, in the Supplementary Information (SI), we introduce a geometric phase diagram with F- and

WF-phase kink widths plotted in terms of $\tilde{\bar{l}} \equiv \bar{l}/a$ and $\tilde{r} \equiv r/a$ (see Supplementary Note 1).

## Computed existence and discrete modal spectra of zero-energy kinks

In this section, we numerically study the existence and discrete modal spectra of F- and WF-phase kinks for varying kink widths (keeping the lattice spacing $a$ fixed). To vary the kink widths, we fix $r$ and vary $\bar{l}$ (equivalent to varying $\bar{\psi}$ in the homogeneous chain). The parameters are arbitrarily chosen as $a = 1$ m, $r = 1.5$ m, $k_e = 100$ N m⁻¹, and $m = 1$ kg, noting that all quantities will be normalized in the subsequent analysis. For each parameter set, the corresponding kink state is obtained numerically using the Newton–Raphson method[64], and its discrete modal spectrum is then determined from the Jacobian matrix (see Supplementary Note 3 for additional details). As mentioned earlier, deep into the F phase ($d \ll 1$), the KL chain can be described by the continuum $\phi^4$ model. To highlight the properties of the KL chain—specifically, the barrier-free propagation and stability of its zero-energy kink—in the SI, we compare them with those of the discrete $\phi^4$ chain[1] (see Supplementary Note 2 for modeling details).

Figure [2]b and c shows the onsite- and intersite-centered KL chain kink states with $N = 280$ rotors and open boundary conditions (insets), together with their corresponding discrete modal spectra as a function of $\widetilde{w}_0$, The shaded regions mark the pass band of the infinite homogeneous chain, and all frequencies are normalized by the lower band edge frequency $\omega_0$ (i.e., $\widetilde{\omega} \equiv \omega/\omega_0$). We first observe that the pass band, as well as the modes within it, collapses at $\widetilde{w}_0 = 0$, during the transition between F and WF phases[10], resulting in a flat dispersion (see Supplementary Note 4 for dispersion relation examples for both KL and $\phi^4$ chains). This can be compared to the $\phi^4$ model as it approaches its, so-called, anti-continuum limit[37]. Supplementary Note 5 provides examples of kink states in discrete $\phi^4$ chains and their corresponding discrete modal spectra as the kink approaches this limit.

In addition to the modes within the pass band, we find additional modes inside the band gap of the infinite homogeneous chain. As expected, the KL chain exhibits a gapped zero mode (with calculated $\widetilde{\omega}^2 < 10^{-7}$) that remains unchanged with kink width for both intersite- and onsite-centered kink states, consistent with the prediction of the Maxwell–Calladine index theorem[12,50,65,66]. This suggests barrier-free kink propagation in the KL chain, regardless of kink width. Short-time dynamic simulations, detailed in Supplementary Note 6, reveal that initiating kink motion in KL chains occurs without a PN barrier, unlike in discrete $\phi^4$ chains, where discreteness-dependent thresholds are observed. We note that both onsite- and intersite-centered F- and WF-phase kinks in the KL chain remain stable due to the zero mode, in contrast to the discrete $\phi^4$ chain, where onsite-centered kink states are unstable[35–38].

In addition to the zero mode, the kink states also exhibit finite-frequency modes within the band gap. These modes are spatially localized around the kink and correspond to the aforementioned internal modes[1,37,38,51]. We find that as the F-phase kink approaches its $\phi^4$ continuum limit ($d \to 0$), only a single internal mode persists, and its frequency asymptotically converges to $\widetilde{\omega}_s^2 = 3/4$, which corresponds to the shape mode (a type of internal mode) of the continuum $\phi^4$ kink[1,37,38,51]. As the WF-phase kink approaches the boundary between the WF and S phases, the onsite-centered kink states exhibit spectra resembling those of the continuum SG model, including the absence of internal modes. Interestingly, towards this WF/S boundary, for the intersite-centered kinks, the lowest frequency internal mode (LFIM) approaches zero, whereas for the onsite-centered kink, the LFIM approaches the upper edge of the band gap. At the boundary between the F and WF phases, we observe that both the onsite- and intersite-centered KL chain's kinks have extra internal modes, compared to those of discrete $\phi^4$ kinks, and all the rest finite-frequency modes collapse into a single frequency, similar to the aforementioned anti-continuum limit of discrete $\phi^4$ kinks. Supplementary Note 7 provides comparison of internal modes in F-phase kinks and discrete $\phi^4$ kinks for a range of kink widths.

We now shift our focus to another key feature of discrete kink-supporting systems. As noted, a general property of these systems, including discrete $\phi^4$ chains, is that they support only two stationary kink states: a stable intersite-centered kink and an unstable onsite-centered kink[35–38]. In contrast, we find that the KL chain supports an infinite set of stationary kink states due to the presence of a zero mode, which, to the best of our knowledge, has never been reported. Here, kink states between two sites are obtained using the transfer and inverse transfer functions of the KL chain, starting from the kink's center rotor angle $\psi_{140}$, and the discrete modes are calculated from solving the eigenproblem of the dynamical matrix for each kink state (see Supplementary Note 8 for details). In Fig. [2]d, we show an example featuring the same kink state that we investigate in our experiments ($\widetilde{r} = 1.5$, $d_{\exp} = 2.0296$), where, as the kink's center shifts between onsite- and intersite-centered, the zero mode consistently exists, showing that the stable kinks can exist anywhere along the KL chain, and the internal modes undergo a continuous and dramatic variation in their frequencies. In Supplementary Note 9, we show that the smoothness of this internal-mode frequency variation depends on $d$.

## Simulated acoustic-wave–kink interaction

In this section, we numerically explore the dynamics of acoustic wave packet interaction with F/WF-phase kinks. The acoustic waves are set to be small-amplitude by ensuring small angular deformations compared to the kink. In Fig. [3], we show the dependence of the acoustic-wave–kink interaction dynamics on the unit cell geometric parameters $d$ and $\widetilde{r}$. In Fig. [4], we show the dependence of acoustic-wave–kink interaction dynamics on wave packet amplitude, wave packet center frequency, and the initial kink's center position, for chains with the same unit cell geometric parameters in our experiments.

Considering first Fig. [3], the top row shows a zoomed-in view of the kink (for the specified, varied, unit cell geometric parameters), the middle row the corresponding pass band of the infinite homogeneous chain, and the bottom row the spatiotemporal chain response in normalized total energy at each site (i.e., the sum of kinetic and potential energy at each site normalized by the maximum in the full spatiotemporal field, denoted by $\widetilde{E}_n$) during the wave packet generation, kink interaction, and subsequent scattering. Time is normalized as $\widetilde{t} \equiv \omega_0 t/(2\pi)$. For these acoustic-wave–kink interaction simulations, we again use the Newton–Raphson method to solve for the onsite-centered kink state comprising the chain's initial configuration, then inject the wave packet by applying a Gaussian-modulated sinusoidal torque $\tau(t) = \tau_0 e^{-(t-t_0)^2/(2\sigma_t^2)} \sin \omega_{\mathrm{driven}} t$ to the left edge (the first rotor), with $t_0 = 40\pi/\omega_{\mathrm{driven}}$, $\sigma_t = 12\pi/\omega_{\mathrm{driven}}$, $\omega_{\mathrm{driven}}$ the excitation frequency centered in the middle of the pass band, and $\widetilde{\tau}_0 \equiv \tau_0/(\omega_0^2 m r^2) = 0.0713$, $0.0021$, $0.0361$, and $0.0171$ for the columns from left to right in Fig. [3], respectively. We denote the normalized displacive amplitude of the wave packet for each case in the bottom left corner of each spatiotemporal diagram. This amplitude is defined as $\widetilde{\delta}_{1,\,\max} \equiv \max(|(\sin\psi_1(t \leq 2t_0) - \sin\psi_{1,t=0})/\sin\bar\psi|)$[10], which represents the first rotor's maximum x-deflection relative to its mass position in the homogeneous chain within $2t_0$ time.

We first describe three cases corresponding to $\widetilde{r} = 1.5$, with $d = 0.3$, 0.95, and 2.0296 (leftmost to third-from-the-left columns in Fig. [3], respectively), wherein the acoustic-wave–kink interaction is attractive, i.e., the kink's velocity is opposite to the velocity of the incoming wave packet. We observe that, when $d$ is small (leftmost column), the kink only moves during the interaction with the wave packet (Fig. [3]c) and there is minimal scattering. This behavior is similar to that of the $\phi^4$ kinks close to their continuum limit[24], which is consistent, as small $d$ approaches the continuum limit that resembles the $\phi^4$ field theory[10]. As $d \to 1$ (second column from the left in Fig. [3]), the kink becomes highly discrete ($\widetilde{\omega}_0 \to 0$), the rotors and springs align more closely, and the pass band narrows, which leads to strong dispersion of the wave packet. As a result, in Fig. [3]f, although initially the kink moves rapidly in the direction of wave packet incidence, it soon slows during interaction with the later-arriving, dispersed waves. We also observe energy propagating away from the kink as it moves, which may constitute a combination of reflection, scattering, and radiation from the kink. For $d > 1$ (third column from the left in Fig. [3]), the kink transitions into a WF-phase kink, where the kink width increases with $d$, but remains relatively small, and the pass band widens, reducing dispersion. After the wave packet interacts with the kink, most of the energy is reflected (unlike the $d \to 0$, continuum limit, case), while a small amount of energy radiates from the kink, mostly opposite to the kink propagation. In addition, we see in the inset of Fig. [3]i that the kink has a somewhat oscillatory trajectory (see Supplementary Note 11 for an enlarged version), which can also be observed in the discrete $\phi^4$ chain (see Supplementary Note 12). The kink also continues to move even after the wave packet moves away from it, again similar to discrete $\phi^4$ kinks[27]. However, the kink dynamics are not identical between the two systems, as the presence of the PN barrier in the discrete $\phi^4$ chain causes the kink to eventually slow and stop due to energy radiating from the kink (see Supplementary Note 12 for a simulated example), an effect not present for the KL chain kinks. Simulation of acoustic-

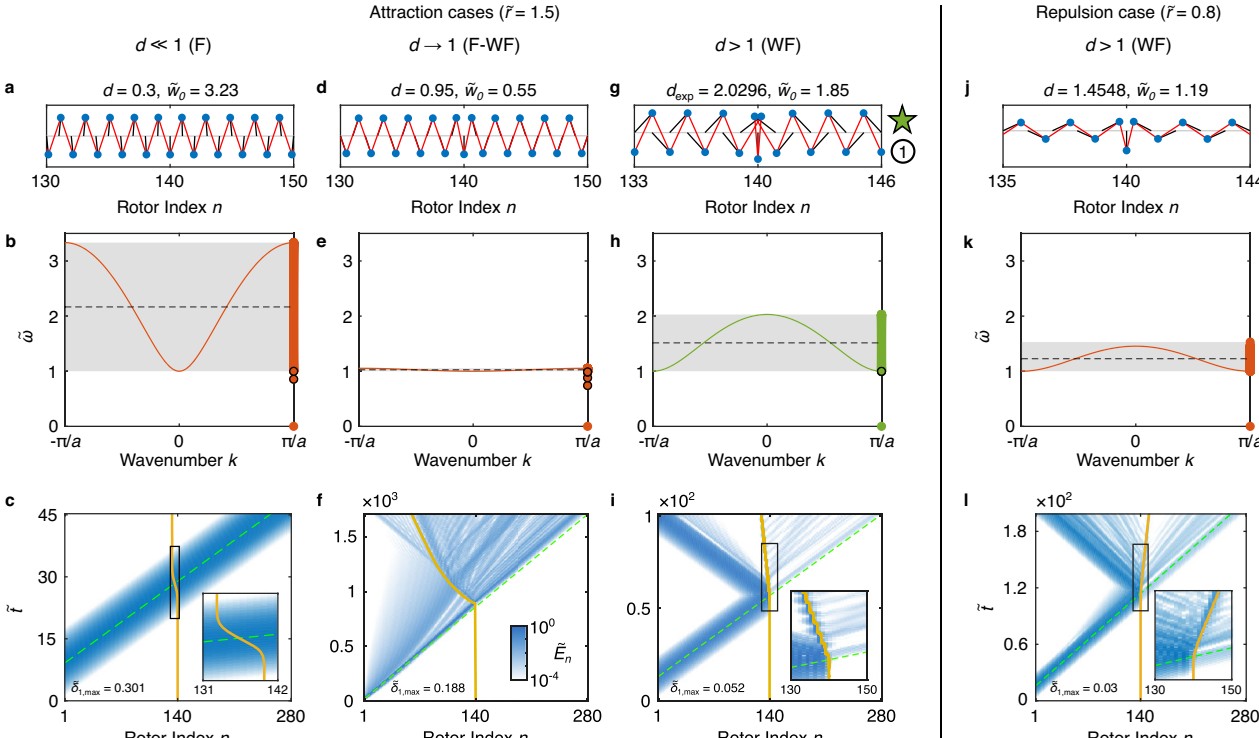

**Fig. 3 | Simulated acoustic-wave-kink interactions in KL chains with different unit cell geometries. a, d, g, j** Zoomed-in view of the onsite-centered kink state for KL chains with 280 rotors. **b, e, h, k** Dispersion relation of infinite homogeneous chain with discrete modes of the finite chain overlapped at $k = \pi/a$ for each case in **a, d, g,** and **j**. The black dashed line denotes the center frequency of the wave packet (middle of the pass band). **c, f, i, l** Spatiotemporal response of normalized total energy at each site (same color bar for all panels), with green dashed lines denoting the predicted wave packet center position based on the group velocity at the excitation frequency in the infinite homogeneous chain and yellow solid lines indicating the fitted center position of the kink (determined by fitting a hyperbolic tangent function to the chain configuration; see Supplementary Note 10 for additional details). The first three columns from the left denote $\tilde{r} = 1.5$ at different $d$ in the F/WF phase, and the rightmost $\tilde{r} = 0.8$ in the WF phase. The circled 1 marker indicates the experimental onsite-centered kink state in Fig. 2.

wave-kink interaction in the $\phi^4$ chain close to its continuum limit is also provided in Supplementary Note 12 for comparison.

By tuning the geometrical parameters ($\tilde{r}$), we observe that the KL chain also supports repulsive interactions between the kink and the wave packet. In the rightmost column of Fig. 3, we show a repulsion case, corresponding to $\tilde{r} = 0.8$ and $d = 1.4548$ (WF-phase). Similar to the attraction case in the third column, we observe that most of the wave packet is reflected after hitting the kink, and the kink continues to move even after the waves have moved away. While kink attraction via acoustic wave interaction is the more common phenomenon[24,27], repulsive behavior during acoustic-wave-kink interaction has also been observed computationally in discrete $\phi^4$[31] and $\phi^6$[43] chains. We also observe stronger radiation as the kink moves, particularly shortly after the interaction. In Supplementary Notes 13 and 14, we provide a first-order perturbation theory calculation that predicts the type of acoustic-wave-kink interaction shortly after the wave packet reaches the kink, as well as numerical exploration of the types of acoustic-wave-kink interaction in KL chains with different unit cell geometries using short-time dynamic simulations.

Noting that the results shown in Fig. 3 are not directly comparable due to the different excitation frequencies and amplitudes involved, as well as the fact that the interaction is a nonlinear process, we next choose the kink state that we will study experimentally and examine (Fig. 4) the influence of wave packet amplitude (top row), wave packet center frequency (middle row), as well as the initial kink's center position (bottom row). The excitation is defined as in Fig. 3. The normalized displacive amplitude of the wave packet is also denoted as in Fig. 3. In Fig. 4a and b, we observe that as the driving torque increases ($\tilde{\tau}_0 = 0.0241$ and $0.0482$), the kink propagates faster and radiates more

in the direction opposite its motion. The kink velocity is extracted from the slope of the traveled rotor index versus time, normalized by the speed of sound (which represents the asymptotic limit of the group velocity at short wavelengths in the near-continuum regime[10]), yielding values of $-0.0165$ and $-0.0581$, respectively. At the highest driving torque ($\tilde{\tau}_0 = 0.0718$), the kink motion becomes random and is accompanied by significant energy radiation (Fig. 4c). As concerns the dependence of the interaction on frequency of the wave packet within the pass band (middle row), with fixed torque amplitude ($\tilde{\tau}_0 = 0.0361$), we observed: differing wave packet dispersal (expected due to differing local band curvature and proximity to band edges), differing levels of kink radiation, and minimally modified (compared to wave packet amplitude effects) kink velocity. For the three cases tested, the kink velocity ranged from $-0.0316$ (Fig. 4d) to $-0.0078$ (Fig. 4f). Finally, as concerns effects of the kink's center position on the acoustic-wave-kink interaction dynamics (bottom row, $\tilde{\tau}_0 = 0.0361$), we observe different amounts of radiation from the kink (following wave packet interaction) and minimal changes in kink velocity ($-0.0441$ to $-0.0534$). We note that while the kink velocities mentioned here may appear small, this is expected because, in the discrete setting, even the maximum group velocity remains below the characteristic sound speed. This is also consistent with the fact that kinks are subsonic solutions whose velocities range from zero up to the sound speed. We suggest this breadth of nonlinear dynamical phenomena demonstrated motivates future studies delving into the mechanisms underlying kink response as a function of wave packet amplitude, wave packet frequency, and kink's center position, as well as their interdependency with the underlying linear dynamical features including the discrete modal spectrum and internal modes.

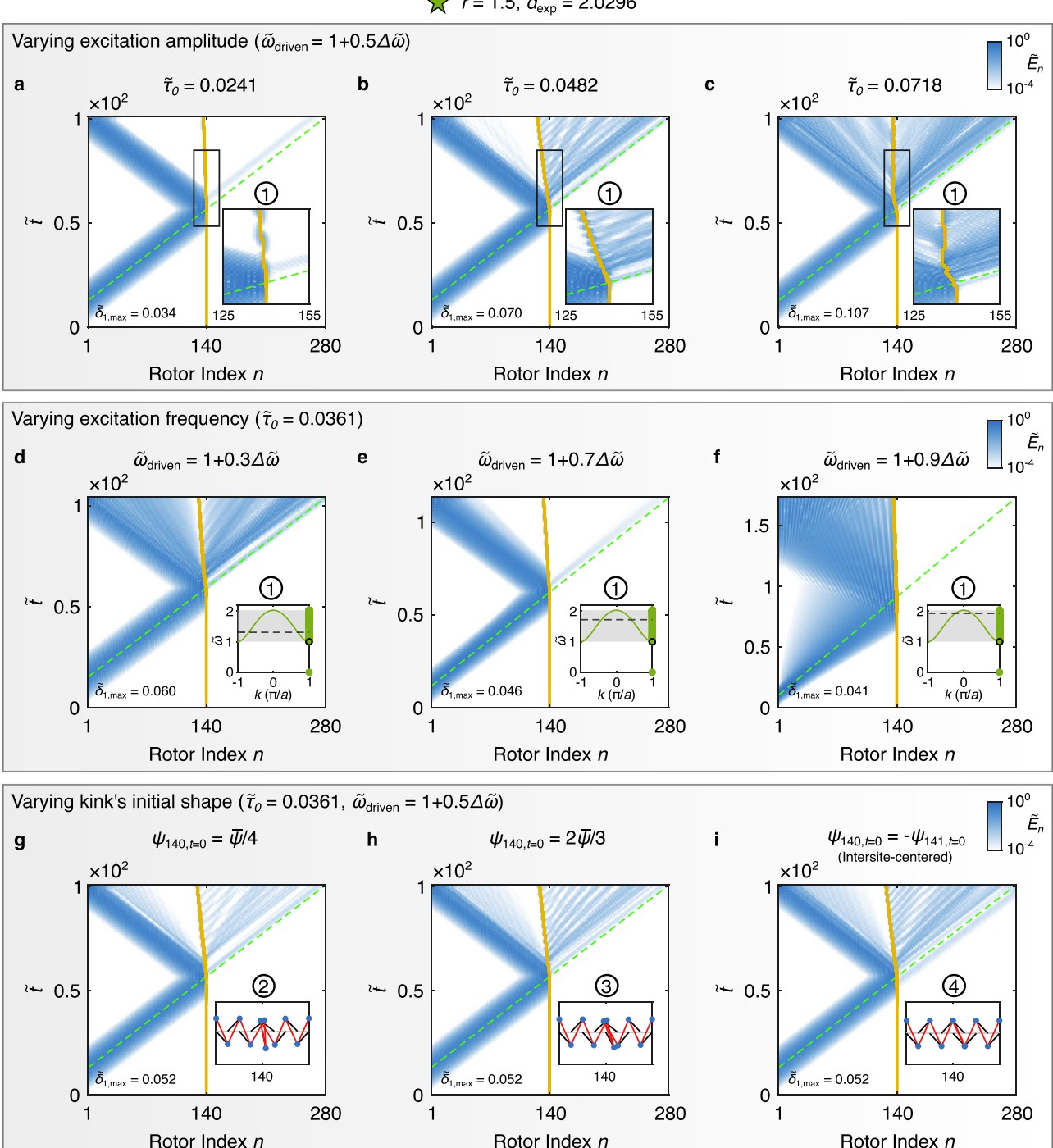

**Fig. 4 | Simulated acoustic-wave–kink interactions in KL chains with the experimental unit cell geometry.** All panels show normalized total energy at each site (same color bar for all panels). **a–c** Kink dynamics under varying excitation amplitudes at a constant mid-band excitation frequency. Insets shows zoomed-in results for the same time window. **d–f** Kink dynamics under different excitation frequencies (black dashed lines in the insets) at fixed excitation torque amplitude. **g–i** Kink dynamics at different kink's center positions, under a constant mid-band excitation frequency and torque amplitude. Circled numbers indicate the kink states with the same number in Fig. 2. The kink state is also shown in the insets. $\Delta\tilde{\omega}$ represents the normalized pass band width, defined as $|\tilde{\omega}_{k=0} - \tilde{\omega}_{k=\pi/a}|$. Green dashed lines denote the predicted wave packet center position based on the group velocity at the excitation frequency in the infinite homogeneous chain and yellow solid lines indicate the fitted center position of the kink.

## Experimental observation of kink control and generation via acoustic waves

To experimentally realize the described KL chain supporting elastic waves, we constructed a rotor chain, in which each rotor was coupled by bent thin polycarbonate beams as effective springs (see Supplementary Note 15 for details). We chose parameters $a = 20$ mm, $r = 30$ mm, and $\bar{l} = 48.5$ mm, which yield $\bar{\psi} = 0.7431$ rad and $d_{exp} = 2.0296$, and place the zero-energy kink in this chain within the WF phase with a small kink width ($\tilde{w}_0 = 1.85$). The springs have ball bearings at the ends, which minimize moments during rotation. The normal stiffness of the spring was measured to be 384.3 N m$^{-1}$. Inspired by the LEGO-based systems in Refs. [10,14], we designed the rotor as shown in Fig. 5a. Red

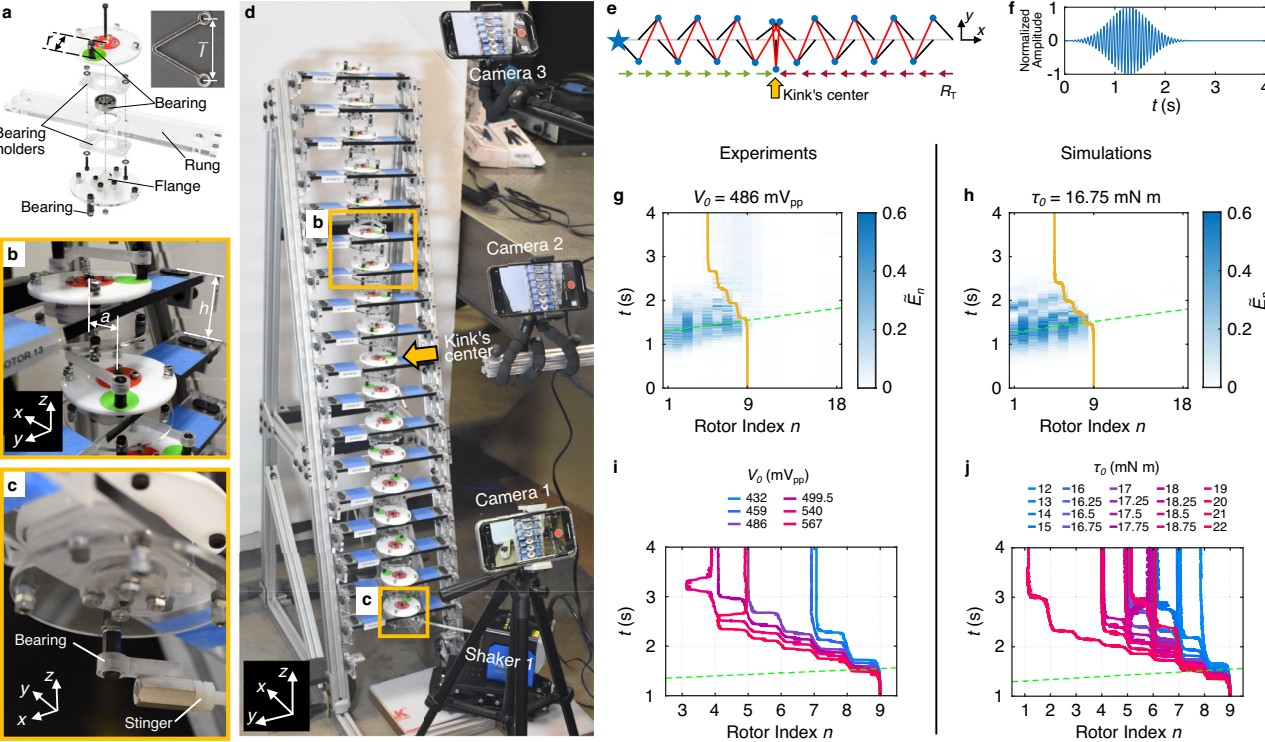

**Fig. 5 | Experimental observation of acoustic-wave–kink interaction (kink control) in the KL chain. a** Single rotor assembly. Inset: a single polycarbonate spring. **b** Two rotors connected via polycarbonate springs. **c** Bottom view of the excited rotor. **d** Experimental setup with chain of 18 rotors configured as in (**e**). **e** Chain with kink centered at the 9th rotor (yellow arrow) with green and red arrows indicating the polarization vectors. The blue star marks the excited rotor ($n = 1$). **f** Normalized excitation function. **g**, **h** Selected experimentally measured (**g**) and simulated (**h**) spatiotemporal chain responses in terms of normalized total energy at each site. Green dashed lines denote the predicted wave packet center position based on the group velocity at the excitation frequency in the infinite homogeneous chain, while yellow solid lines indicate the fitted kink's center position. Color bars are truncated for clarity. **i**, **j** Fitted kink's center trajectories for different excitation amplitudes in experiments (**i**) and simulations (**j**). The driving voltages amplitudes in (**i**) were randomly chosen, with the upper and lower limits chosen based on when increased interplay with the excitation boundary and minimal kink motion was observed, respectively. The driving torque amplitudes in (**j**) were chosen based on a tri-linear equal spacing of torques, with the most dense spacing in the vicinity of 17.5 mN m.

and green markers indicate the disc's center and the spring's bearing, respectively. Mass was evenly distributed to avoid wobble and the moment of inertia of an entire rotor is $2.97 \times 10^{-5}$ kg m$^2$ (see Supplementary Note 16 for details). The central ball bearing of the rotor was installed on a horizontal rung. By staggering rungs with a relative out-of-plane ($z$) distance ($h = 55$ mm, Fig. 5b), overlapping configurations of the KL chain were enabled. Figure 5d shows the experimental KL chain consisting of 18 rotors.

To initialize the configuration with a kink in the middle of the chain, we first set the rotor angles to those of the homogeneous state, then formed and moved the kink from the soft edge by applying large manual deformations until $\psi_9 = 0$ (Fig. 5e). To inject acoustic wave packets into the chain, an electrodynamic shaker (shaker 1 in Fig. 5d) was coupled to the bottom edge of the chain (the first rotor) via a ball bearing and a nylon stinger, which provided a tangential linear excitation (Fig. 5c). The excitation voltage signal applied to the shaker (with built-in amplifier) followed the same Gaussian-modulated sinusoidal torque used in simulations (Fig. 5f), with $\omega_{driven} = 2\pi f$ and $f = 15.65$ Hz (the median of the eigenfrequencies calculated from the chain configuration in Fig. 5e). See Supplementary Note 17 for details on the excitation signal setting for both experiment and simulation. We tracked the dynamic response of the chain using digital image processing of synchronized videos recorded at 240 fps by three iPhones, approximately equidistantly spaced along the chain (Fig. 5d; see Supplementary Note 18 for additional details on experimental setup and data acquisition).

In Fig. 5g the kink can be seen to move upon interaction with the wave packet. In contrast to the simulated continuous kink motion

(see Supplementary Note 19 also for simulated demonstration of this continuous motion in a longer chain with the same parameters as in experiment), we observed that the kink stops moving after about four sites. We attribute this to damping, which, due to the lack of a PN barrier, becomes the dominant mechanism resisting kink motion in this system. In many other examples, such as dislocation motion in crystals[67], while damping may be present, the PN barrier plays the primary role in resisting kink motion. In Fig. 5g and h, we compare our experimental result with a simulated result (selected based on similar propagation distance as the experiment), now with the inclusion of an experimentally determined onsite viscous damping (see Supplementary Note 20 for details). We note the excitation implementation differs between the two cases: in the experiments, a linear shaker applied tangential excitation to the rotor, which constrains its motion; in simulation, the excitation was modeled as a torque applied directly to the rotor, allowing free rotation. To verify that the motion of the kink is not due to gravity (since it moves downward), we excited the top edge of the chain (18th rotor) and observed that the wave packets indeed attract the kink (see Supplementary Movies 1 and 2 for demonstrations of exciting bottom and top edge, respectively). In addition, we excited both ends of the chain at a frequency in the band gap (5 Hz) and did not observe any movement of the kink as expected (see Supplementary Movies 3 and 4). To visualize the contribution of internal modes, we further performed a mode decomposition of the results shown in Fig. 5g and h. Specifically, we decomposed the acoustic-wave–kink interaction displacements onto the eigenmodes of the initial kink configuration (see Supplementary Note 21 for computational details

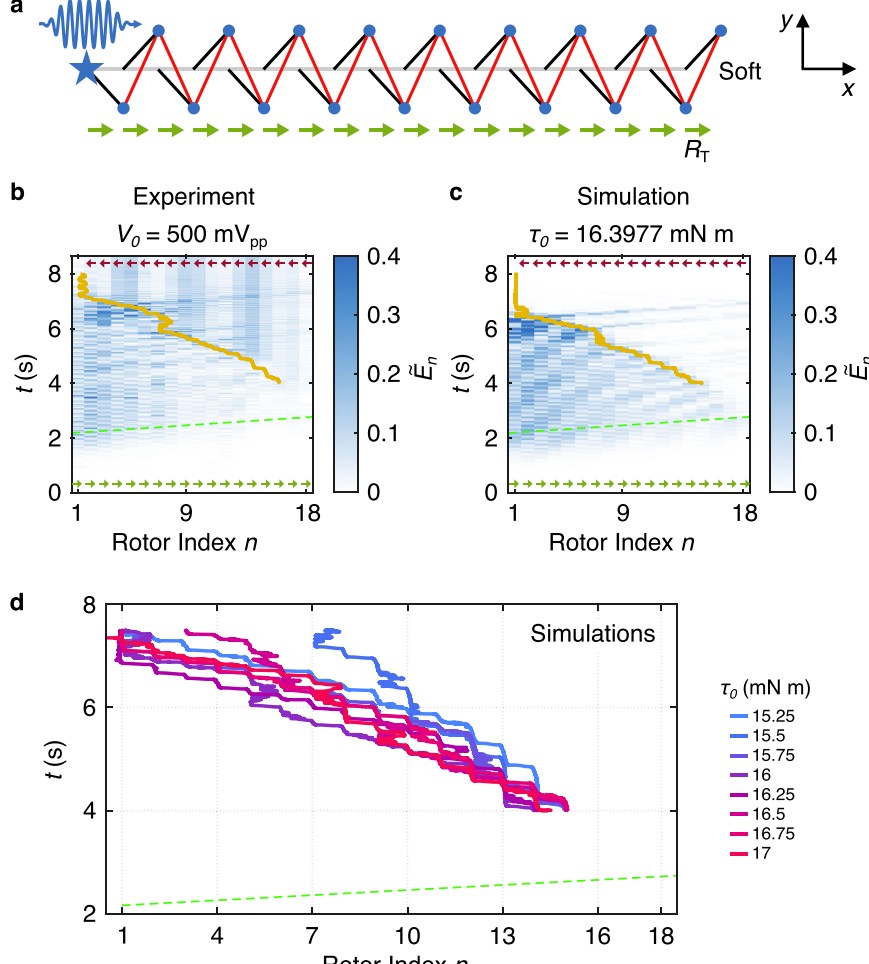

**Fig. 6 | Experimental observation of kink generation via acoustic wave packets in the KL chain. a** Homogeneous chain configuration with green arrows indicating the initial polarization vectors (RPS). The blue star marks the excited rotor ($n = 1$). **b**, **c** Example experimentally measured (**b**) and simulated (**c**) spatiotemporal chain response in terms of normalized total energy at each site, with green dashed lines denoting the group velocity at the excitation frequency in the infinite homogeneous chain (with start time corresponding to the end of the ramp) and yellow solid lines indicating the kink's fitted center position from 4 s to 8 s. Color bars are truncated for visualization purposes. Small arrows in (**b**) and (**c**) indicate the initial (green) and final (red) polarization vectors. **d** Fitted kink's center trajectories for different values of excitation amplitudes in simulations, with the kink's center position fitted from 4 s to 7.5 s.

and results), showing internal modes are excited shortly after the wave packet interaction.

To demonstrate the ability to control kink propagation via acoustic-wave–kink interaction, we examined the influence of various excitation amplitudes. The resulting kink trajectories from experiments are shown in Fig. 5i. We observed that higher excitation amplitudes generally enhance both the propagation distance and speed of the kink, as further illustrated by simulations in Fig. 5j (see Supplementary Note 22 for spatiotemporal responses for both experiments and simulations). At larger amplitudes, we also observed instances of backward kink motion during propagation, in both experiments and simulations. We attributed this to a combination of boundary effects (due to the short chain length), the close proximity of the excitation source, and the high excitation amplitudes. In Supplementary Note 23, we isolated and examined the individual contributions of these factors by simulating both the experimental chain and its extended versions. We also provide additional simulations showing the control over kink propagation by adjusting the excitation frequency, excitation width, and initial kink's center position, as well as the onsite damping.

In Fig. 6, we show that a kink can be generated from the soft edge of an initially homogeneous chain by injecting acoustic wave packets from the opposite, rigid edge. Although kink generation has been

demonstrated in active systems[45–47], we show it can be realized with only passive elements. Exciting the same experimental chain in the homogeneous state (Fig. 6a) with a longer, ramped sinusoid (12.7 Hz frequency, residing in the pass band), we observed the formation and propagation of a kink, switching the polarization of the bulk, in both experiment (Fig. 6b) and simulation (Fig. 6c, including damping, and selected based on similar kink trajectory as in Fig. 6b). See Supplementary Notes 17 and 22 for details on the excitation signal setting and spatiotemporal response for both experiment and simulation. Again, we observed that most of the waves were reflected by the kink. In Fig. 6d, we numerically examine the influence of varied excitation amplitude. Although the kink trajectories vary slightly among these cases, a consistent feature is that the kink does not propagate at a perfectly constant velocity. This non-monotonic behavior can be attributed to the combined effects of long-duration driving and the close proximity of the boundary, which together lead to multiple interactions and reflections. To examine the kink dynamics under negligible boundary effects, we also simulated the same scenarios in an analogous, longer chain (see Supplementary Note 24), wherein the kink traveled a less distance due to the minimized acoustic wave reflections from the boundaries. In that longer chain, the kink still propagated farther at higher excitation amplitudes, though its motion became increasingly non-monotonic. We also provided additional

experimental tests with excitation outside the pass band (6.35 Hz and 25.4 Hz), where no kink generation was observed (see Supplementary Movies 5–7 for these tests). In Supplementary Note 25, we further simulated a repulsion case of the kink in the WF phase for parameters corresponding to the experiment, both with and without onsite damping. While the repulsion can be observed for the case without damping for these parameters, we were unable to observe it for the case with damping. Increasing the driving amplitude for the damped case in an attempt to observe such repulsion, resulted in the formation of a chaotic dynamical state before repulsion was seen. Because of this, we did not attempt to experimentally observe repulsion.

## Discussion

In this work, we reported the first experimental observation of mechanical kink control and generation via acoustic wave packets, including, in particular, highly discrete kinks. This was enabled by the use of an elastically-coupled KL chain, supporting zero-energy kinks. Future related research directions include extending our system into higher dimensions and smaller scales, as well as answering fundamental questions regarding kink dynamics in the KL chain. Regarding higher dimensions, several works have studied the existence of domain walls in 2D topological metamaterials that support zero-energy modes[57,68]. Regarding scalability, we anticipate possibilities in scale reduction, consider prior demonstration of rotor-like behaviors in systems such as bacterial flagellar motors[60], DNA[61], and nanoelectromechanics[62]. Given the importance of highly discrete kinks in the form of dislocations in condensed matter[1], one might ask, what are the thermal properties of materials supporting highly discrete kinks that lack a PN barrier? Several near-term, future questions regarding kink dynamics in the KL chain are also stimulated by our results. What are the dynamics of the antikink in the KL chain[11,14], noting that they are not zero-energy and have a finite PN barrier? This includes their interaction with moving zero-energy kinks as well as acoustic-wave–antikink interaction. What are kink dynamics in a quasi-periodic KL chain[69]? Is there a relationship between the internal mode evolution with the kink's center position between sites and the short-time attraction and repulsive behavior? Given these applied and fundamental possibilities, we anticipate fruitful extensions of discreteness-independent, barrier-free acoustic-wave–kink interaction dynamics. Finally, we suggest our findings may lead to analogs in quantum regimes for controlling kinks with negligible PN barriers via true phonons, including the incorporation of phenomena unique to such scales (*e.g.*, tunnel nucleation of kinks[70]).

## Data availability

The datasets that support the findings of this study are available in the Zenodo repository under https://doi.org/10.5281/zenodo.18028283.

## Code availability

The codes that generate the numerical findings of this study are available in the Zenodo repository under https://doi.org/10.5281/zenodo.18039150.

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

## Acknowledgements

K.Q., N.C., X.M., and N.B. acknowledge support from the US Army Research Office (Grant No. W911NF-20-2-0182). N.C., F.S., K.S., and X.M. acknowledge support from the US Office of Naval Research (MURI N00014-20-1-2479). This research is funded in part by a grant from ICAM the Institute for Complex Adaptive Matter to K.Q. In addition, K.Q. acknowledges support from the UCSD MAE Stanford S. Penner Post-Doctoral Research Travel Award. N.H. acknowledges support from the UC President's Dissertation Year Fellowship.

## Author contributions

K.Q. performed theoretical and numerical analyses, designed the experiments, and built the experimental setup. K.Q. and N.H. conducted the experiments. N.C. and F.S. carried out additional theoretical analyses and simulations. K.S., G.T., X.M., and N.B. supervised the work. All authors discussed the results and contributed to the manuscript.

## Competing interests

The authors declare no competing interests.
