## [Transparent Peer Review file · Nature Communications]

Observation of mechanical kink control and generation via acoustic waves

Corresponding Author: Professor Nicholas Boechler

Version 0:

Reviewer comments:

Reviewer #1

(Remarks to the Author)

The manuscript by Qian et al. presents a comprehensive investigation into the generation and control of mechanical kinks via mechanical vibrations (for which the authors and others use the term 'phonons') in a topological metamaterial, specifically a Kane-Lubensky (KL) chain. The study effectively combines theoretical modeling, numerical simulations, and experimental realization to explore phonon-kink interactions in a system that supports zero-energy kinks, circumventing the traditional Peierls–Nabarro (PN) barrier. The work builds on well-established concepts such as topologically protected solitons and the phase classification of kink modes (Flipper, Wobbling Flipper, Spinner), and extends them through a careful examination of dynamical properties in both damped and undamped systems. While the experimental demonstration of phonon-induced kink movement is a meaningful step forward, that could fit dissemination in Nature Communications, several aspects of the work would benefit from additional clarification, especially concerning the transition from simulation to experiment and the identification of novel contributions. We list specific comments and questions here below.

1. The central experimental result — demonstrating kink generation and control via phonons — is intriguing and constitutes a noteworthy advancement. However, while Figures 3 and 4 showcase a rich variety of simulated phonon-kink interaction types, including directional propagation, speed variations, and attraction/repulsion regimes, these nuanced dynamics appear absent from the experimental data. Could the authors clarify whether any experimental control over kink velocity, direction, or dynamic state was observed, and if so, provide supporting data or extended measurements? Specifically, is it possible to control the kink propagation speed or other dynamic behaviors by altering the geometry or excitation conditions?
2. In Figure 3, the authors discuss different types of interactions between phonons and kinks based on variations in unit cell geometry. Is there a well-defined phase boundary separating the different interaction regimes, particularly between attractive and repulsive cases?
3. Much of the theoretical and structural foundation of the work—including the zero-energy kink in the KL chain, the absence of the PN barrier, and the classification into Flipper, Wobbling Flipper, and Spinner phases—is well established in the literature and extensively cited (e.g., Refs. [11] and [12]). Figures 1 and 2 are well-composed but largely recapitulate known properties. Could the authors clarify how their current results build upon or go beyond these foundational works? For instance, are the internal mode families or smooth mode transitions shown in Fig. 2e previously unreported?
4. In Figure 5, the authors mention that damping suppresses kink propagation. Without reducing the damping itself, is it possible to extend the propagation distance by modifying the geometry or the nature of the excitations?
5. In Figure 6, the authors show that the kink does not propagate at a constant speed in either experiment or simulation, and there appears to be a significant disturbance around unit cells 6–7. What is the origin of this behavior in the simulations? Could it be due to the finite system size or boundary effects?
6. Following the previous point, in the simulation without damping (Fig.S20), the kink propagates over a long distance but remains far from the boundaries due to the large system size. Could the authors clarify how they separate the impact of damping from possible finite-size or boundary-related effects?

Reviewer #2

(Remarks to the Author)

The manuscript of Qian et al. models and experimentally observes the interactions of phonons with kinks in the Kane-Lubensky chain. The authors show that the gapless nature of KL kinks renders them particularly easy to steer via small amplitude wave packets, because there is no Peierls-Nabarro discreteness barrier to overcome. They argue that control of gapless kinks, exemplified by the KL chain, represent a promising paradigm for reconfigurable materials.

In short, I believe this is interesting work and I recommend its publication in Nature Communications.

Control of solitons in mechanical metamaterials is an area of substantial current interest within mechanical engineering and soft matter physics. This paper advances the art by providing an initial study of how kinks in KL chains can be manipulated, with a corresponding experimental observation.

That said, there are several aspects of the manuscript I believe could be clarified, and I offer concrete suggestions for improvements, alongside some questions, below:

- (1) The authors argue that the serious impediment to kink control is the PN barrier. However, another problem which the authors also encounter is dissipation. Could the authors comment on the relative importance of these two problems – why is the PN barrier really the critical issue to overcome, as opposed to damping of phonon packages and kink motion?
- (2) Could the authors comment on the efficiency of the phonon driving? For example, if I take Fig. 3c, most of the driving energy passes straight through the kink. Are there strategies to optimize the coupling between the driving packet and the kink motion?
- (3) The experimental size of the KL chain seems highly limited by the need to Z-stack to avoid self-contact. Could the authors comments on the prospect of experimentally scaling these systems to useful dimensions?
- (4) The authors make many comparisons between the KL chain and the discrete ϕ^4 dynamics. As the authors mention, another limit of the KL chain is sine-Gordon. Could the authors comments on why ϕ^4 , as opposed to sine-Gordon, is the right comparison to make?
- (5) Several aspects of Fig. 2 are hard for me to follow: Could the authors explain in more detail exactly what the “phonon spectrum” shown in Fig2 (c,f etc) is? How is it calculated? My interpretation is that first a nonlinear kink solution is numerically found using Newton Raphson. Next, the eigenmodes of the dynamics linearized about that kink state define this phonon spectrum. For me, the main text doesn't explain this explicitly enough. Fig. 2b and Fig. 2e are also hard to follow, and the caption of Fig. 2 jumps between figures somewhat out of order. I think Fig 2b is showing a 1D curve through 3D configuration space, coloured by both d and w_0 . But the aspect ratio of the 3D plot makes this hard to follow. For Fig. 2e, I am not entirely sure what is being shown, and the text is unclear. Perhaps a schematic figure would help here.

More Minor comments:

- (6) The units are hard to follow. For example, what does a kink velocity is -0.0885 (other examples across pages 6-7) mean? Is that big or small? I appreciate that these are simulation units, but some context for what this effectively dimensionless velocity means would be helpful.

Reviewer #3

(Remarks to the Author)

The authors show, numerically and experimentally, that travelling wave packets ('phonons') can be used to control (move around, create) kinks in a Kane-Lubensky type mechanical metamaterial. This primary result is supported by extensive numerical simulations, investigating stability, phase diagrams, effect of damping, etc.

I think this is a very nice work and I recommend accepting with very minor suggestions (see later). It is very nice that a topological defect can be shuttled, in the lab, in a controllable way around a material, and I can see how this could lead to exciting applications (shaping proteins, computing, etc). It is also pretty nice that there are regimes of attractive and repulsive interactions, although it seems that the repulsive regime is a bit harder to achieve in practice.

The manuscript is well-written and the methodology is sound in both theory and experiments, adequately supporting the claims. I did not see any important detail missing to reproduce the work.

Minor comments:

- (a) In page 5 the authors mention 'a small amount of phonons'. I'm pretty sure they do not have a small amount of phonons in such macroscopic, low-frequency experiment. I would expect the number of phonons to be on the order of 10^{25} . Perhaps the authors could drop or reduce the word 'phonons' there, or at least be clear that they are operating in a classical regime, and in no case should 'small number of phonons' be used in this setup.

- (b) [optional, would require larger edits] The paper does a significant amount of theory (stability analysis, phase diagrams, simulations, etc) before demonstrating the experimental control of kinks with phonons (Fig. 5) and the generation of kinks (Fig. 6). I find the stability analysis and so on good to support the key hypothesis. However, since phonon effects on boundary defects have been predicted elsewhere (including in some of the references), perhaps the manuscript would send a clearer message if the experiments were more prominent and the stability analysis and so on provided as extended data.

Version 1:

Reviewer comments:

Reviewer #1

(Remarks to the Author)

I have reviewed the response and changes made by the authors. They have significantly improved their manuscript, which I now recommend for publication.

Reviewer #2

(Remarks to the Author)

The authors have provided a substantial amount of work to address my comments. I repeat my recommendation for publication in Nature Communications.

Reviewer #3

(Remarks to the Author)

The authors have addressed (most) of my concerns, and I think most of the concerns from the other reviewers. I'm particularly impressed that they manage to experimentally demonstrate most of the soliton phenomena that they had predicted theoretically---even though I would not have expected it was necessary for publication.

I still struggle a bit with the fact that the authors now redefine the word phonon.

For 100 years, the word phonon has referred to a unit of energy in a specific mode (ie. $\hbar\omega$, which for the author's system would be c.a. 10^{-30} J). Phonons do not look like wavepackets, they look like standing waves --- although you can combine them to form a wavepacket, but then they are longer a pure phonon. A person with a traditional physics training, when said that 'phonons' are put into the system, will think that the system is oscillating with a standing wave with amplitudes around 10^{-20} meters.

If the authors want to say that they put wavepackets, why don't call them wavepackets? If long-term, quantum applications of this idea are on the horizon (which is fine), it is OK to add a paragraph in the outlook that maybe one day this could be made with single phonons and this is an area to be studied.

Response to Referees. Referee comments are in blue text. Replies to referee comments are in black text. Revised text incorporated into the updated manuscript is in red text. Copies of the main text and SM with red text denoting the revisions are included as an additional, single pdf file.

I. REPLY TO REFEREE 1

“The manuscript by Qian et al. presents a comprehensive investigation into the generation and control of mechanical kinks via mechanical vibrations (for which the authors and others use the term ‘phonons’) in a topological metamaterial, specifically a Kane-Lubensky (KL) chain. The study effectively combines theoretical modeling, numerical simulations, and experimental realization to explore phonon-kink interactions in a system that supports zero-energy kinks, circumventing the traditional Peierls–Nabarro (PN) barrier. The work builds on well-established concepts such as topologically protected solitons and the phase classification of kink modes (Flipper, Wobbling Flipper, Spinner), and extends them through a careful examination of dynamical properties in both damped and undamped systems. While the experimental demonstration of phonon-induced kink movement is a meaningful step forward, that could fit dissemination in Nature Communications, several aspects of the work would benefit from additional clarification, especially concerning the transition from simulation to experiment and the identification of novel contributions. We list specific comments and questions here below.”

We thank the referee for the effort and time they put in reviewing our manuscript. Below we respond to each question raised by the referee.

“1. The central experimental result — demonstrating kink generation and control via phonons — is intriguing and constitutes a noteworthy advancement. However, while Figures 3 and 4 showcase a rich variety of simulated phonon-kink interaction types, including directional propagation, speed variations, and attraction/repulsion regimes, these nuanced dynamics appear absent from the experimental data. Could the authors clarify whether any experimental control over kink velocity, direction, or dynamic state was observed, and if so, provide supporting data or extended measurements? Specifically, is it possible to control the kink propagation speed or other dynamic behaviors by altering the geometry or excitation conditions?”

To experimentally validate the capacity for deterministic control of kink propagation via mechanical vibrations (phonons), we performed additional experiments focusing on the effect of vibration amplitude, while keeping the frequency and width of the wave packet fixed, in the same experimental chain. The resulting kink trajectories demonstrate that, in general, stronger excitations increase both the distance traveled and the velocity of the kink. We also found this to be in agreement with additional numerical simulations that we conducted and have added to the revised manuscript.

Regarding the predicted repulsive regime, we did not attempt to observe this as noted in the original manuscript: “While the repulsion can be observed for the case without damping for these parameters, we were unable to observe it for the case with damping. Increasing the driving amplitude for the damped case in an attempt to observe such repulsion, resulted in the formation of a chaotic dynamical state before repulsion was seen. Because of this, we did not attempt to experimentally observe repulsion.” Two other control knobs that showed strong effects in simulation were excitation frequency and kink phase and differing unit cell geometries. The effects of placing the excitation frequency in the band gap were shown in the SM and mentioned in the main text of the original manuscript: “We also excited both ends of the chain at a frequency in the band gap (5 Hz) and did not observe any movement of the kink as expected (see the SM Video 3 and 4 for exciting bottom and top edge, respectively)”. However, we did not experimentally test the effects of changing the driving frequency within the band gap. We also did not experimentally test differing unit cell geometries. We plan to leave experimental investigation of these two knobs for future work.

We have included the new experimental results and their corresponding simulations, and updated the related paragraphs in both the revised main text and Supplementary Materials (SM). Additional simulations for kink control in both the experimental chain and its extended version are provided in the revised SM. Additions are as follows.

Main text: *“To demonstrate the ability to control kink propagation via phonon-kink interaction, we examined the influence of various excitation amplitudes. The resulting kink trajectories from experiments are shown in Fig. 5(i). We observed that higher excitation amplitudes generally enhance both the propagation distance and speed of the kink, as further illustrated by simulations in Fig. 5(j). At larger amplitudes, we also observed instances of backward kink motion during propagation, in both experiments and simulations. We attributed this to a combination of boundary*

effects (due to the short chain length), the close proximity of the excitation source, and the high excitation amplitudes. In the SM, we isolated and examined the individual contributions of these factors by simulating both the experimental chain and its extended versions. We also provide additional simulations showing the control over kink propagation by adjusting the excitation frequency, excitation width, and initial kink center position, as well as the onsite damping.”

FIG. R1. FIG. 5. **Experimental observation of phonon-kink interaction (kink control) in the KL chain.** (a) Single rotor assembly. Inset: a single polycarbonate spring. (b) Two rotors connected via polycarbonate springs. (c) Bottom view of the excited rotor. (d) Experimental setup with chain of 18 rotors configured as in (e). (e) Chain with kink centered at the 9th rotor (yellow arrow) with green and red arrows indicating the polarization vectors. The blue star marks the excited rotor ($n = 1$). (f) Normalized excitation function. (g,h) Selected experimentally measured (g) and simulated (h) spatiotemporal chain responses in terms of normalized total energy at each site. Green dashed lines denote the predicted wave packet center position based on the group velocity at the excitation frequency in the infinite homogeneous chain, while yellow solid lines indicate the fitted kink center position. Color bars are truncated for clarity. (i,j) Fitted kink center trajectories for different excitation amplitudes in experiments (i) and simulations (j). The driving voltages amplitudes in (i) were randomly chosen, with the upper and lower limits chosen based on when increased interplay with the excitation boundary and minimal kink motion was observed, respectively. The driving torque amplitudes in (j) were chosen based a tri-linear equal spacing of torques, with the most dense spacing in the vicinity of 17.5 mN·m.

SM:

XXIV. ADDITIONAL EXPERIMENTAL RESULTS AND COMPARISON WITH SIMULATIONS

A. Kink control with phonons: Tuning wave packet amplitude

To demonstrate kink control by varying wave packet amplitude, Figs. R2, R4, R6, and R8 show, for different excitation amplitudes, our experimental KL chain’s measured spatiotemporal response in terms of angular position, normalized angular velocity, spring strain, and total energy per rotor, respectively. These spatiotemporal responses correspond to the cases shown in Fig. 5(i). A few simulated example responses with different excitation amplitudes are provided in Figs. R3, R5, R7, and R9 for comparison (corresponding to Fig. 5(j)). As can be seen in the main text (Fig. 5(i,j)), in both experiment and simulation, we see that, in general, stronger excitations increase both the distance traveled and the velocity of the kink.

FIG. R2. FIG. S30. Experimentally measured spatiotemporal chain responses (corresponding to Fig. 5(i) in the main text) in terms of angular position for kink control cases at various excitation amplitudes, with green dashed line denoting the predicted wave packet center position based on the group velocity at the excitation frequency in the infinite homogeneous chain.

FIG. R3. FIG. S31. Six simulated spatiotemporal chain responses (at 1 mN·m intervals selected from Fig. 6(j) in the main text) in terms of angular position for kink control cases at various excitation amplitudes, with green dashed line denoting the predicted wave packet center position based on the group velocity at the excitation frequency in the infinite homogeneous chain.

FIG. R4. FIG. S32. Experimentally measured spatiotemporal chain responses (corresponding to Fig. 5(i) in the main text) in terms of normalized angular velocity for kink control cases at various excitation amplitudes, with green dashed line denoting the predicted wave packet center position based on the group velocity at the excitation frequency in the infinite homogeneous chain and yellow solid line indicating the fitted position of the kink center.

FIG. R5. FIG. S33. Six simulated spatiotemporal chain responses (at 1 mN·m intervals selected from Fig. 6(j) in the main text) in terms of normalized angular velocity for kink control cases at various excitation amplitudes, with green dashed line denoting the predicted wave packet center position based on the group velocity at the excitation frequency in the infinite homogeneous chain and yellow solid line indicating the fitted position of the kink center.

FIG. R6. FIG. S34. Experimentally measured spatiotemporal chain responses (corresponding to Fig. 5(i) in the main text) in terms of spring strain for kink control cases at various excitation amplitudes, with green dashed line denoting the predicted wave packet center position based on the group velocity at the excitation frequency in the infinite homogeneous chain and yellow solid line indicating the fitted position of the kink center. $l_{n,n+1}^{(t=0)}$ denotes the length of the spring connecting rotors n and $n+1$ at $t=0$, calculated from the tracked rotor angles in the recorded frame.

FIG. R7. FIG. S35. Six simulated spatiotemporal chain responses (at 1 mN·m intervals selected from Fig. 6(j) in the main text) in terms of spring strain for kink control cases at various excitation amplitudes, with green dashed line denoting the predicted wave packet center position based on the group velocity at the excitation frequency in the infinite homogeneous chain and yellow solid line indicating the fitted position of the kink center.

FIG. R8. FIG. S36. Experimentally measured spatiotemporal chain responses (corresponding to Fig. 5(i) in the main text) in terms of normalized total energy at each site for kink control cases at various excitation amplitudes, with green dashed line denoting the predicted wave packet center position based on the group velocity at the excitation frequency in the infinite homogeneous chain and yellow solid line indicating the fitted position of the kink center. Color bar is truncated for visualization purposes.

FIG. R9. FIG. S37. Six simulated spatiotemporal chain responses (at 1 mN·m intervals selected from Fig. 6(j) in the main text) in terms of normalized total energy at each site for kink control cases at various excitation amplitudes, with green dashed line denoting the predicted wave packet center position based on the group velocity at the excitation frequency in the infinite homogeneous chain and yellow solid line indicating the fitted position of the kink center. Color bar is truncated for visualization purposes.

A. Kink generation with phonons

Figures R10 and R11 show the experimental and simulated spatiotemporal chain responses in terms of angular position, angular velocity, and spring strain, respectively, for the kink generation case as shown in Fig. 6.

FIG. R10. FIG. S38. Experimentally measured spatiotemporal chain responses in terms of (a) angular position, (b) normalized angular velocity, and (c) spring strain for the kink generation case, with green dashed lines denoting the group velocity at the excitation frequency in the infinite homogeneous chain (with start time corresponding to the end of the ramp) and yellow solid lines indicating the kink's fitted center position from 4 s to 8 s. $l_{n,n+1}^{(t=0)}$ denotes the length of the spring connecting rotors n and $n + 1$ at $t = 0$, calculated from the tracked rotor angles in the recorded frame.

FIG. R11. FIG. S39. Simulated spatiotemporal chain responses in terms of (a) angular position, (b) normalized angular velocity, and (c) spring strain for the kink generation case, with green dashed lines denoting the group velocity at the excitation frequency in the infinite homogeneous chain (with start time corresponding to the end of the ramp) and yellow solid lines indicating the kink's fitted center position from 4 s to 8 s.

“2. In Figure 3, the authors discuss different types of interactions between phonons and kinks based on variations in unit cell geometry. Is there a well-defined phase boundary separating the different interaction regimes, particularly between attractive and repulsive cases?”

We have not yet been able to analytically identify a well-defined phase boundary that separates the attractive and repulsive phonon-kink interaction regimes. However, to address this point, we performed short-time dynamic simulations to visualize the transition between different interaction types. These simulations were conducted on an undamped KL chain consisting of 100 rotors, with the kink centered at the middle of the chain. Phonons were injected from the first rotor in the form of a Gaussian-modulated sine wave packet at the midband frequency for each case. We fitted the kink velocity within a short time window during the wave packet interaction to obtain the kink acceleration, whose sign indicates the interaction type. The results (Fig. R12) show that while most geometries exhibit attraction-type interactions, repulsive interactions emerge near the lower boundary of the geometric phase diagram.

We have added this phase diagram to the SM and revised the main text to reference it.

Main text: “In the SM, we provide a first-order perturbation theory calculation that predicts the type of phonon-kink interaction shortly after the wave packet reaches the kink, as well as numerical exploration of the types of phonon-kink interaction in KL chains with different unit cell geometries using short-time dynamic simulations.”

SM:

XV. NUMERICAL INVESTIGATION OF DIFFERENT PHONON-KINK INTERACTION REGIMES FOR THE ZERO-ENERGY KINKS IN THE KL CHAIN

We numerically investigated the types of phonon-kink interactions in KL chains with different unit cell geometries using short-time dynamic simulations. As in the previous section, the parameters were set as $a = 1$ m, $k_e = 100$ N/m, $m = 1$ kg, and $N = 100$, with the kink centered at rotor $N/2 = 50$ ($\psi_{50} = 0$). Phonons were injected through the same torque applied to the first rotor as described in the previous section. Each simulation was run for a total duration of $t_0 + Na/v_g$, where v_g is the group velocity of phonons in the corresponding homogeneous chain. The simulations were carried out in MATLAB using the *ode45* solver, with ‘RelTol’ and ‘AbsTol’ both set to 10^{-6} . By linearly fitting the angular velocity data of rotor 50 ($\dot{\psi}_{50}$) between t_1 and t_2 (defined in the previous section), we obtained the corresponding kink accelerations, which are overlaid on the geometric phase diagram, as shown in Fig. S19.

For most unit cell geometries across all three phases, the phonon-kink interaction exhibits an attraction-type behavior (negative acceleration), which tends to become stronger as \tilde{r} and \tilde{l} decrease. Along the lower boundary of the diagram, however, the interaction transitions to a repulsive type (positive acceleration) and gradually weakens toward the WF-S phase boundary. The transition between the two interaction regimes does not appear smooth, and the boundary separating them is clearly distinguishable. A weakly repulsive case is also observed in the WF phase, away from the boundary.

“3. Much of the theoretical and structural foundation of the work—including the zero-energy kink in the KL chain, the absence of the PN barrier, and the classification into Flipper, Wobbling Flipper, and Spinner phases—is well established in the literature and extensively cited (e.g., Refs. [11] and [12]). Figures 1 and 2 are well-composed but largely recapitulate known properties. Could the authors clarify how their current results build upon or go beyond these foundational works? For instance, are the internal mode families or smooth mode transitions shown in Fig. 2e previously unreported?”

The purpose of Fig. 1 is purely to illustrate the central idea of our work, which is moving the static zero-energy kink in the KL chain through phonon-kink interaction. To our knowledge, this has not been demonstrated before. To further highlight the experimental contribution of this work, we added the schematic diagram in Fig. 1 (Fig. R13) with an accompanying section of the experimental chain.

Regarding Fig. 2, to better emphasize our main results, we have revised Fig. 2 (see Fig. R14 and accompanying revised text, below). As part of this, we have moved the geometric phase diagram, the short-time dynamic simulations, and the discrete ϕ^4 kink results, along with related paragraphs, to the SM.

FIG. R12. FIG. S19. Phonon-kink interaction types in KL chains with different unit cell geometries from short-time dynamic simulations. Red and blue colors indicate fitted kink acceleration (negative: attraction, positive: repulsion). Green star indicates the experimental kink state.

Contrasting then the differences between Refs. [11,12] and the content in the revised Fig. 2, we note the following. In Fig. 2(a), we show the phase classification of the KL chain and the dependence of the kink width on the phase index d . This is essentially the same as what was shown in Ref. [11], but we believe it is important for setting the stage for our subsequent results. Past this, Ref. [12] numerically compared the discrete modal frequencies of the discrete chain with continuum predictions and confirmed the vanishing translational mode for all kinks considered. However, their discussion of internal modes were restricted to the lowest frequency internal (shape) mode and covered only a limited range of kink widths, without addressing the highly discrete (narrow) kink configurations. *In Fig. 2 and related text, we showed that additional internal modes emerge within the band gap of the kink spectrum, particularly for kink widths smaller than 1, which has not been reported before. In addition, the smooth mode transitions as a function of the kink center position that we shown in Fig. 2 has not been reported before.*

To illustrate the importance of these internal modes, in the revised manuscript, we added a mode decomposition of the spatiotemporal chain response in terms of angular displacement, which reveals the short-time energy distribution across eigenmodes during phonon-kink interaction in the KL chain. In this case, both experiment and simulation show that energy initially populates the zero mode and rapidly transfers to the internal and other phonon modes.

In support of the revision of Fig. 2, we have also added the modal decomposition results in the SM and revised the main text to clarify how our current results build upon and extend previous foundational work. Our revisions concerning these points are as follows.

Main text:

Main text: “Figure 2(b,c) show the onsite- and intersite-centered KL chain kink states with $N = 280$ rotors and open boundary conditions (insets), together with their corresponding discrete modal spectra as a function of \tilde{w}_0 . The shaded regions mark the phonon band ranges of the infinite homogeneous chain, and all frequencies are normalized by the lower band edge frequency ω_0 (*i.e.*, $\tilde{\omega} \equiv \omega/\omega_0$). We first observe that the phonon band, as well as the modes within it, collapses at $\tilde{w}_0 = 0$, during the transition between F and WF phases [1], resulting in a flat dispersion (see the SM for dispersion relation examples for both KL and ϕ^4 chain). This can be compared to the ϕ^4 model as it approaches its, so-called, anti-continuum limit [2]. The SM provides examples of kink states in discrete ϕ^4 chains and their corresponding discrete modal spectra as the kink approaches this limit.

In addition to the modes within the phonon band, we find additional modes inside the band gap of the infinite

FIG. R13. **FIG. 1. Overview of a phonon wave packet moving the static zero-energy kink in the KL chain.** (a) Schematic of a chain section, where black lines represent massless rotors with radius r and lattice constant a , blue circles represent point masses m , and red lines represent linear normal springs with spring constant k_e and instantaneous length $l_{n,n+1}$ (with n and $n + 1$ as rotor indices), accompanied by a photograph of the corresponding experimental chain section. (b) Kink state before (top) and after (bottom) phonon wave packet interaction. The yellow dashed circle indicates the center of the kink.

homogeneous chain. As expected, the KL chain exhibits a gapped zero mode (with calculated $\tilde{\omega}^2 < 10^{-7}$) that remains unchanged with kink width for both intersite- and onsite-centered kink states, consistent with the prediction of the Maxwell-Calladine index theorem [3–6]. This suggests “barrier-free” kink propagation in the KL chain, regardless of kink width. Short-time dynamic simulations, detailed in the SM, reveal that initiating kink motion in KL chains occurs without a PN barrier, unlike in discrete ϕ^4 chains, where discreteness-dependent thresholds are observed. We note that both onsite- and intersite-centered F- and WF-phase kinks in the KL chain remain stable due to the zero mode, in contrast to the discrete ϕ^4 chain, where onsite-centered kink states are unstable [2, 7–9].

In addition to the zero mode, the kink states also exhibit finite-frequency modes within the band gap. These modes are spatially localized around the kink and correspond to the aforementioned internal modes [2, 9–11]. We find that as the F-phase kink approaches its ϕ^4 continuum limit ($d \rightarrow 0$), only a single internal mode persists, and its frequency asymptotically converges to $\tilde{\omega}_s^2 = 3/4$, which corresponds to the shape mode (a type of internal mode) of the continuum ϕ^4 kink [2, 9–11]. As the WF-phase kink approaches the boundary between the WF and S phases, the onsite-centered kink states exhibit spectra resembling those of the continuum SG model, including the absence of internal modes. Interestingly, towards this WF/S boundary, for the intersite-centered kinks, the lowest frequency internal mode (LFIM) approaches zero, whereas for the onsite-centered kink, the LFIM approaches the upper edge of the band gap. At the boundary between the F and WF phases, we observe that both the onsite- and intersite-centered KL chain’s kinks have extra internal modes, compared to those of discrete ϕ^4 kinks, and all the rest finite-frequency modes collapse into a single frequency, similar to the aforementioned anti-continuum limit of discrete ϕ^4 kinks. The SM provides comparison of internal modes in F-phase kinks and discrete ϕ^4 kinks for a range of kink widths.

We now shift our focus to another key feature of discrete kink-supporting systems. As noted, a general property of these systems, including discrete ϕ^4 chains, is that they support only two stationary kink states: a stable intersite-centered kink and an unstable onsite-centered kink [2, 7–9]. In contrast, we find that the KL chain supports an infinite set of stationary kink states due to the presence of a zero mode, which, to the best of our knowledge, has never been reported. Here, kink states between two sites are obtained using the transfer and inverse transfer functions of the KL chain, starting from the kink’s center rotor angle ψ_{140} , and the discrete modes are calculated from solving the eigenproblem of the dynamical matrix for each kink state (see SM for details). In Fig. 2(d), we show an example featuring the same kink state that we investigate in our experiments ($\tilde{r} = 1.5$, $d_{\text{exp}} = 2.0296$), where, as the kink’s center shifts between onsite- and intersite-centered, the zero mode consistently exists, showing that the stable kinks can exist anywhere along the KL chain, and the internal modes undergo a continuous and dramatic variation in their frequencies. In the SM, we show that the smoothness of this internal-mode frequency variation depends on d .”

FIG. R14. FIG. 2. **Computed discrete modal spectra of KL chain's F/WF-phase kinks, including a family of internal modes with smooth variation between sites.** (a) Phase and kink width diagram of the zero-energy kink in the KL chain as a function of the dimensionless index d [1]. (b,c) Normalized eigenfrequencies of kink states as a function of their normalized widths \tilde{w}_0 for KL chains with $\tilde{r} = 1.5$ and $N = 280$ rotors, where (b) and (c) show results for onsite- ($\psi_{140} = 0$) and intersite-centered ($\psi_{140} = -\psi_{141}$) kinks, respectively. Markers with black edge indicate internal modes. Insets show zoomed-in, normalized kink states ($-\sin\psi_n/\sin\bar{\psi}$ [1]). Green color denotes the kink state examined in experiments, which is further used in (d), as well as Figs. 3 and 4, to denote the same geometry. For F- and WF-phase kinks, the lowest eigenfrequency remains zero, indicating barrier-free propagation independent of kink width (*i.e.*, discretization). (d) In-gap modes of the experimental kink state as a function of the kink's center rotor angle. ① and ④ indicate onsite- and intersite-centered kinks, respectively, while ② and ③ correspond to intermediate kink states between them, which are $\psi_{140}/\bar{\psi} = 1/4$ and $2/3$ (further used in Figs. 3 and 4).

SM:

XXV. MODE DECOMPOSITION FOR SHORT-TIME PHONON-KINK INTERACTION IN KINK CONTROL

To study the short-time energy distribution across eigenmodes during the phonon-kink interaction, we projected the experimental and simulated displacement results of the kink control case shown in Fig. 5(g,h) onto the eigenvector basis of the initial chain configuration (with a kink onsite-centered at the middle of the chain), obtained using the Newton-Raphson method [12]. The results are shown in Fig. S40, with time windows shifted between experimental and simulated plots for visual alignment. Mode 1 corresponds to the zero mode, while Mode 2 (10.41 Hz) and Mode 3 (10.48 Hz) correspond to the internal modes (indicated by the red box). The excitation frequency (15.65 Hz) is between the frequencies of Mode 9 (15.08 Hz) and Mode 10 (16.21 Hz).

We observe that, in both experiment and its corresponding simulation, the energy initially distributes into the zero mode and then rapidly transfers to the internal modes and the other phonon modes. We note that the time window is slightly different between the experiment and simulation cases (Fig. S40(a) and (b)). This shifted window

shown was chosen to center the time when the modal weight of the first mode (the zero mode) begins increasing in amplitude. We suggest that this difference in zero mode excitation time between experiment and simulation may be due to differences in group velocity between reality and our model. In our results, we note that the excitation and the interaction predominantly involve odd-index phonon modes rather than even-index ones.

FIG. R15. FIG. S40. **Mode decomposition results for short-time phonon-kink interaction in the kink control case shown in Fig. 5(g,h).** Mode frequencies increase with mode index. Green dashed lines mark the predicted arrival time of the wave packet center at the kink center position and red boxes indicate the internal modes.

“4. In Figure 5, the authors mention that damping suppresses kink propagation. Without reducing the damping itself, is it possible to extend the propagation distance by modifying the geometry or the nature of the excitations?”

During the revision of this manuscript, we performed new experiments using the same configuration (thus no change in damping). By keeping the same frequency and duration of the injected phonon wave packet, we showed that the kink propagation distance and velocity can indeed be controlled by varying the excitation amplitude. Both experiments and simulations demonstrate that the kink propagation distance and velocity generally increase with increasing excitation amplitude, up to a certain limit.

In addition, we performed further simulations of the phonon–kink interaction in both the experimental chain (18 rotors) and an extended chain (280 rotors), keeping the same damping coefficient. These results demonstrate that, without reducing damping, the kink propagation distance can be further extended by modifying either the excitation parameters (*e.g.*, amplitude, frequency, and width) or the chain geometry (*e.g.*, initial kink position and chain length).

In summary, while damping indeed suppresses kink propagation, our results show that it is still possible to enhance the propagation distance through appropriate adjustments of the excitation characteristics or the system geometry.

The new experimental results can be seen in the updated Fig. 5(i) in the main text, as well as in the updated SM (Figs. S30, S32, S34, and S36). Additional simulations (in addition to the results shown in Fig. 5(j) in the main text and Figs. S31, S33, S35, and S37) have been included in the SM, as follows.

SM:

XXVII. ADDITIONAL SIMULATIONS OF KINK CONTROL IN THE KL CHAIN WITH EXPERIMENTAL PARAMETERS AND ONSITE DAMPING

A. Simulations of varying excitation amplitude

We performed numerical studies on the experimental ($N = 18$) and extended ($N = 280$) KL chains, both incorporating onsite damping ($c_{exp} = 1.274 \times 10^{-4}$ N·m·s/rad), to examine the effect of varying excitation amplitude. The kink was placed at the chain center ($\psi_{N/2} = 0$) and the first rotor was driven according to Eq. S98.

Figure S43 presents the simulated spatiotemporal response of the 18-rotor chain under varying driving torque amplitudes, including additional cases with finer amplitude steps near that shown in Fig. 5(h). Together with Fig. S37, the results show that increasing the excitation amplitude generally enhances the kink's propagation distance, velocity, and associated energy radiation. Beyond a certain threshold (*e.g.*, the last panel in Fig. S43), however, the propagation distance decreases, and the kink motion can become less uniform or even reverses direction, as seen for $\tau_0 = 18$ mN·m in Fig. S37. These counterintuitive behaviors are attributed to boundary effects, where the kink interacts with waves reflected from the chain ends.

FIG. R16. FIG. S43. Simulated spatiotemporal responses in terms of normalized total energy at each site of the 18-rotor KL chain with onsite damping under varying driving torque amplitudes. Green dashed lines denote the predicted wave packet center position based on the group velocity at the excitation frequency in the infinite homogeneous chain and yellow solid lines indicate the fitted position of the kink center. Color bars are truncated for visualization purposes.

To minimize boundary effects, we simulated a 280-rotor KL chain using the same unit-cell parameters and damping values as before. As shown in Fig. S44, the kink propagates a shorter distance at the same excitation amplitude compared with the 18-rotor chain, confirming that boundary reflections contribute to the extended propagation observed previously. Similar to the shorter chain, increasing the excitation amplitude enhances kink motion and velocity up to a threshold, which shifts to a higher amplitude in the longer system. At sufficiently high amplitudes, chaotic responses and even backward propagation emerge. Overall, these results demonstrate that kink dynamics can

be effectively tuned through the excitation amplitude.

FIG. R17. FIG. S44. Simulated spatiotemporal responses in terms of normalized total energy at each site of the 280-rotor KL chain with onsite damping under varying driving torque amplitudes. Green dashed lines denote the predicted wave packet center position based on the group velocity at the excitation frequency in the infinite homogeneous chain and yellow solid lines indicate the fitted position of the kink center. Color bars are truncated for visualization purposes.

B. Simulations of varying excitation frequency

We examined the effect of varying excitation frequency (within the phonon band) in the 18- and 280-rotor KL chains, both with the same onsite damping, excitation setup, and kink placement as before.

Figure S45 shows the simulated spatiotemporal response of the 18-rotor chain under different driving torque frequencies. The results indicate that varying the excitation frequency can cause substantial changes in the kink dynamics and the associated energy radiation. These variations are again attributed to boundary effects arising from the short chain length. Figure S46 shows that once boundary effects become negligible, the kink propagates a shorter distance, consistent with the previous amplitude-dependent cases. Moreover, varying the excitation frequency results in distinct propagation behaviors, confirming that kink motion can be controlled by tuning the excitation frequency.”

FIG. R18. FIG. S45. Simulated spatiotemporal responses in terms of normalized total energy at each site of the 18-rotor KL chain with onsite damping under varying driving torque frequencies. Green dashed lines denote the predicted wave packet center position based on the group velocity at the excitation frequency in the infinite homogeneous chain and yellow solid lines indicate the fitted position of the kink center. Color bars are truncated for visualization purposes.

FIG. R19. FIG. S46. Simulated spatiotemporal responses in terms of normalized total energy at each site of the 280-rotor KL chain with onsite damping under varying driving torque frequencies. Green dashed lines denote the predicted wave packet center position based on the group velocity at the excitation frequency in the infinite homogeneous chain and yellow solid lines indicate the fitted position of the kink center. Color bars are truncated for visualization purposes.

C. Simulations of varying excitation width

We examined the effect of varying excitation width (*i.e.*, time duration) in the 18- and 280-rotor KL chains, both with the same onsite damping, excitation setup, and kink placement as before.

Figures S47 and S48 show the simulated spatiotemporal response of the 18- and 280-rotor chain under different driving torque widths ($\sigma_t = 0.15N_{cycle}/f$; the shifted starting time of the predicted wave packet center position is due to $t_0 = N_{cycle}/(2f)$), respectively. We observed that increasing the excitation width generally enhances the kink's propagation distance and velocity, as well as the associated energy radiation. These results confirm that kink motion can be effectively controlled by tuning the excitation width.

FIG. R20. FIG. S47. Simulated spatiotemporal responses in terms of normalized total energy at each site of the 18-rotor KL chain with onsite damping under varying driving torque widths. Green dashed lines denote the predicted wave packet center position based on the group velocity at the excitation frequency in the infinite homogeneous chain and yellow solid lines indicate the fitted position of the kink center. Color bars are truncated for visualization purposes.

FIG. R21. FIG. S48. Simulated spatiotemporal responses in terms of normalized total energy at each site of the 280-rotor KL chain with onsite damping under varying driving torque widths. Green dashed lines denote the predicted wave packet center position based on the group velocity at the excitation frequency in the infinite homogeneous chain and yellow solid lines indicate the fitted position of the kink center. Color bars are truncated for visualization purposes.

D. Simulations of varying initial kink center positions

We examined the effect of varying the initial kink center position in the 18- and 280-rotor KL chains, both with the same onsite damping and excitation setup as before.

In the 18-rotor chain (Fig. S49, the kink's mobility decreases as it is positioned farther from the excitation source, which can be attributed to reduced boundary influence and phonon dissipation over longer propagation distances. In the 280-rotor chain (Fig. S50, where boundary effects are negligible, the kink's mobility still decreases with distance from the excitation source, isolating the role of damping in the system. Furthermore, Fig. S51 shows that when boundary effects are absent and phonon dissipation remains comparable (*i.e.*, the kink is placed the same distance from the excitation source), the kink and phonon dynamics change only minimally. Overall, these results confirm that tuning the initial kink position influences propagation only when boundary effects are present.

FIG. R22. FIG. S49. Simulated spatiotemporal responses in terms of normalized total energy at each site of the 18-rotor KL chain with onsite damping and different initial kink center positions. The excitation is applied at rotor 1. Green dashed lines denote the predicted wave packet center position based on the group velocity at the excitation frequency in the infinite homogeneous chain and yellow solid lines indicate the fitted position of the kink center. Color bars are truncated for visualization purposes.

Varying initial kink center position ($N = 280$)
 $(f = 15.65 \text{ Hz}, \tau_0 = 23 \text{ mN}\cdot\text{m}, N_{\text{cycle}} = 40)$

FIG. R23. FIG. S50. Simulated spatiotemporal responses in terms of normalized total energy at each site of the 280-rotor KL chain with onsite damping and different initial kink center positions. The excitation is applied at rotor 132. Green dashed lines denote the predicted wave packet center position based on the group velocity at the excitation frequency in the infinite homogeneous chain and yellow solid lines indicate the fitted position of the kink center. Color bars are truncated for visualization purposes.

FIG. R24. FIG. S51. Simulated spatiotemporal responses in terms of normalized total energy at each site of the 280-rotor KL chain with onsite damping and different initial kink center positions. The excitation is applied eight rotors to the left of the kink center. Green dashed lines denote the predicted wave packet center position based on the group velocity at the excitation frequency in the infinite homogeneous chain and yellow solid lines indicate the fitted position of the kink center. Color bars are truncated for visualization purposes.

SM:

F. Simulations of varying chain lengths

We examined the effect of varying the chain length of the KL chain. The excitation was applied eight rotors to the left of the kink center. Figure S54 shows that increasing the chain length reduces boundary effects, as evidenced by the decrease in kink mobility.

FIG. R25. FIG. S54. Simulated spatiotemporal responses in terms of normalized total energy at each site of the KL chain with onsite damping and different chain lengths. The excitation is applied eight rotors to the left of the kink center. Green dashed lines denote the predicted wave packet center position based on the group velocity at the excitation frequency in the infinite homogeneous chain and yellow solid lines indicate the fitted position of the kink center. Color bars are truncated for visualization purposes.

“5. In Figure 6, the authors show that the kink does not propagate at a constant speed in either experiment or simulation, and there appears to be a significant disturbance around unit cells 6–7. What is the origin of this behavior in the simulations? Could it be due to the finite system size or boundary effects?”

The additional kink generation simulations that we conducted (varying amplitude and chain length for the same chain geometry) and added to the revised manuscript show that the kink generally does not propagate at a constant velocity. In addition, this change in velocity happens at a range of unit cells (instead of just unit cells 6-7).

Regarding the potential cause of the non-constant velocity, our newly-added simulations of a 280-rotor chain (keeping the same source-to-boundary distance as in the shorter chain) show that the non-constant velocity appears reduced at low amplitudes compared to the shorter chain. The kink does still exhibit nonuniform motion at larger excitation amplitudes. This longer chain simulation reduces contributions from one of the boundaries. These results suggest that nonuniform behaviors stem from a combination of nonlinear effects occurring at large excitation amplitude and the close proximity between the source, kink, and boundary.

These points have been clarified via revisions to the main text and additions to the SM, as follows.

Main text: “In Fig. 6(d), we numerically examine the influence of varied excitation amplitude. Although the kink trajectories vary slightly among these cases, a consistent feature is that the kink does not propagate at a perfectly constant velocity. This non-monotonic behavior can be attributed to the combined effects of long-duration driving and the close proximity of the boundary, which together lead to multiple interactions and reflections. To examine the kink dynamics under negligible boundary effects, we also simulated the same scenarios in an analogous, longer chain (see SM), wherein the kink traveled a less distance due to the minimized phonon reflections from the boundaries. In that longer chain, the kink still propagated farther at higher excitation amplitudes, though its motion became increasingly non-monotonic.”

FIG. R26. **FIG. 6. Experimental observation of kink generation via phonons in the KL chain.** (a) Homogeneous chain configuration with green arrows indicating the initial polarization vectors (RPS). The blue star marks the excited rotor ($n = 1$). (b,c) **Example** experimentally measured (b) and simulated (c) spatiotemporal chain response in terms of normalized total energy at each site, with green dashed lines denoting the group velocity at the excitation frequency in the infinite homogeneous chain (with start time corresponding to the end of the ramp) and yellow solid lines indicating the kink’s fitted center position from 4 s to 8 s. Color bars are truncated for visualization purposes. Small arrows in (b,c) indicate the initial (green) and final (red) polarization vectors. (d) Fitted kink center trajectories for different values of excitation amplitudes in simulations, with the kink’s center position fitted from 4 s to 7.5 s.

SM:

XXVIII. ADDITIONAL SIMULATIONS OF KINK GENERATION VIA PHONONS IN THE KL CHAIN WITH EXPERIMENTAL PARAMETERS AND ONSITE DAMPING

We performed numerical studies on the experimental ($N = 18$) and extended ($N = 280$) KL chains, both in the right-polarized state (RPS) and incorporating onsite damping, to examine the effect of varying excitation amplitude. The first rotor was driven according to Eq. S100.

Figure S55 presents additional simulations for the 18-rotor RPS chain with finer amplitude steps around the representative case shown in Fig. 6. Nonuniform kink motion is observed in all these cases.

FIG. R27. FIG. S55. Simulated spatiotemporal responses in terms of normalized total energy at each site of the 18-rotor RPS KL chain with onsite damping under different excitation amplitudes. The excitation is applied at rotor 1. Green dashed lines denote the group velocity at the excitation frequency in the infinite homogeneous chain (with start time corresponding to the end of the ramp) and yellow solid lines indicate the kink's fitted center position from 4 s to 8 s (4 s to 7.5 s for the last panel). Color bars are truncated for visualization purposes.

In the 280-rotor chain (Fig. S56), the kink propagates a shorter distance at the same excitation amplitude, confirming that boundary effects contribute to the enhanced mobility observed in the shorter chain. At higher amplitudes (last row), the kink travels farther but exhibits increasingly nonuniform motion, suggesting that this behavior arises not only from boundary effects but also from strong excitation, close proximity, and prolonged phonon–kink interactions, with minor influence from reflections at the right boundary. Overall, these results demonstrate that kink generation and subsequent motion can be effectively tuned by adjusting the excitation amplitude.

FIG. R28. FIG. S56. Simulated spatiotemporal responses in terms of normalized total energy at each site of the 280-rotor RPS KL chain with onsite damping under different excitation amplitudes. The excitation is applied at rotor 263. Green dashed lines denote the group velocity at the excitation frequency in the infinite homogeneous chain (with start time corresponding to the end of the ramp) and yellow solid lines indicate the kink’s fitted center position from 4 s to 8 s. Color bars are truncated for visualization purposes.

“6. Following the previous point, in the simulation without damping Fig.S20), the kink propagates over a long distance but remains far from the boundaries due to the large system size. Could the authors clarify how they separate the impact of damping from possible finite-size or boundary-related effects?”

In simulations for the experimental chain (18 rotors) that we have added to the revised SM, we observed that reducing damping can either enhance or weaken kink mobility. We attribute this counterintuitive behavior to boundary effects inherent to the short chain, where phonons reflected from the boundaries reinteract with the kink. To disentangle the effects of damping from finite-size effects, we performed additional simulations using the same geometry but with an extended 280-rotor chain. In this case, where boundary effects are negligible, reducing damping consistently enhances kink mobility (both propagation distance and velocity). This comparison allows us to isolate the role of damping independently of boundary-induced phenomena.

The aforementioned simulation results included in the revised SM are as follows.

SM:

E. Simulations of varying onsite damping

We examined the effect of varying the onsite damping in the 18- and 280-rotor KL chains, both with the same excitation setup and kink placement as before.

In the 18-rotor chain (Fig. S52), reducing damping can either enhance or weaken kink mobility, a counterintuitive behavior that we again attribute to boundary effects arising from the short chain length. In the 280-rotor chain (Fig. S53), where boundary effects are negligible, reducing damping enhances kink mobility. This demonstrates that kink propagation can be effectively controlled by tuning the intrinsic damping of the chain.

FIG. R29. FIG. S52. Simulated spatiotemporal responses in terms of normalized total energy at each site of the 18-rotor KL chain with different onsite damping values. Green dashed lines denote the predicted wave packet center position based on the group velocity at the excitation frequency in the infinite homogeneous chain and yellow solid lines indicate the fitted position of the kink center. Color bars are truncated for visualization purposes.

FIG. R30. FIG. S53. Simulated spatiotemporal responses in terms of normalized total energy at each site of the 280-rotor KL chain with different onsite damping values. Green dashed lines denote the predicted wave packet center position based on the group velocity at the excitation frequency in the infinite homogeneous chain and yellow solid lines indicate the fitted position of the kink center. Color bars are truncated for visualization purposes.

II. REPLY TO REFEREE 2

“The manuscript of Qian et al. models and experimentally observes the interactions of phonons with kinks in the Kane-Lubensky chain. The authors show that the gapless nature of KL kinks renders them particularly easy to

steer via small amplitude wave packets, because there is no Peierls-Nabarro discreteness barrier to overcome. They argue that control of gapless kinks, exemplified by the KL chain, represent a promising paradigm for reconfigurable materials.”

“In short, I believe this is interesting work and I recommend its publication in Nature Communications.”

“Control of solitons in mechanical metamaterials is an area of substantial current interest within mechanical engineering and soft matter physics. This paper advances the art by providing an initial study of how kinks in KL chains can be manipulated, with a corresponding experimental observation.”

“That said, there are several aspects of the manuscript I believe could be clarified, and I offer concrete suggestions for improvements, alongside some questions, below:”

We appreciate very much the positive comments from the referee. Below we respond to each question raised by the referee.

“(1) The authors argue that the serious impediment to kink control is the PN barrier. However, another problem which the authors also encounter is dissipation. Could the authors comment on the relative importance of these two problems – why is the PN barrier really the critical issue to overcome, as opposed to damping of phonon packages and kink motion?”

The Peierls–Nabarro (PN) barrier, which arises from the discrete nature of the system and sets the minimum energy required for a kink to propagate. A higher PN barrier also enhances phonon radiation, causing the kink to lose energy and eventually come to rest. The presence of damping further accelerates this process by continuously removing energy from the system, in addition to the energy lost via phonon radiation, causing the kink to stop even sooner than it would due to the PN barrier alone. In some systems damping may indeed be the dominant mechanism resisting kink motion. However, in many other examples, such as dislocation motion in crystals [13], the PN barrier is the dominant factor.

Main Text: **“We attribute this to damping, which, due to the lack of a PN barrier, becomes the dominant mechanism resisting kink motion in this system. In many other examples, such as dislocation motion in crystals [13], while damping may be present, the PN barrier plays the primary role in resisting kink motion.”**

“(2) Could the authors comment on the efficiency of the phonon driving? For example, if I take Fig. 3c, most of the driving energy passes straight through the kink. Are there strategies to optimize the coupling between the driving packet and the kink motion?”

To quantify the efficiency of phonon driving, we performed two simulations of an attraction-type interaction under distinct excitation amplitudes and calculated the corresponding energy transmission from an input wave packet to a moving kink. The normalized kink velocities after the interaction (relative to the speed of sound [1]) are -0.0137 and -0.0759 , corresponding to input torques of $\tilde{\tau}_0 = 0.006647$ and 0.02216 , respectively. Approximately 2.27% and 6.35% of the phonon energy are transferred to the kink in each respective case, while the remaining energy is transmitted, reflected, or radiated away. This quantification has been added to the SM as follows.

SM:

XVI. EXAMPLES OF ENERGY TRANSMISSION FROM AN INPUT PHONON WAVE PACKET TO A MOVING KINK THROUGH PHONON-KINK INTERACTION IN THE KL CHAIN

To illustrate the efficiency of phonon driving, we present two examples of energy transmission from an input phonon wave packet to a moving kink through phonon–kink interaction in the KL chain. We selected the kink state with parameters $\tilde{r} = 0.9$ and $\tilde{l} = 1.35$, which exhibits an attraction-type interaction, as shown in Fig. S19. The phonon–kink interaction was simulated in a 280-rotor chain, where the first rotor was driven at the mid-band frequency with two different driving amplitudes ($\tilde{\tau}_0 = 0.006647$ and 0.02216). The corresponding spatiotemporal responses are shown in Fig. S20. The normalized kink velocities (relative to the speed of sound [1]) after the interaction are fitted as -0.0137 and -0.0759 , respectively. In Fig. S20, the red dash-dotted line indicates the time $t = 3t_0$ (when the entire phonon wave packet has entered the chain) used to quantify the phonon energy, which corresponds to the total chain energy

since the kink is initially static and has zero energy before the interaction. The blue dash-dotted line marks the time $t = Na/v_g$, used to evaluate the kink energy. Although the normalized kink width \tilde{w}_0 is approximately 1.3, we selected the kink center rotor (around rotor 137 and 120, respectively), together with its three neighboring rotors on each side (seven rotors in total), at $t = Na/v_g$ to compute the total kink energy. By comparing the moving-kink energy with the input phonon wave packet energy, we find that approximately 2.27% of the phonon energy is transferred to the kink in (a) and about 6.35% in (b), while the remaining energy is transmitted, reflected, or radiated away from the kink during and after the interaction as the kink propagates.

FIG. R31. FIG. S20. Simulated spatiotemporal chain responses in terms of normalized total energy at each site of phonon-kink interaction for a kink state with parameters $\tilde{r} = 0.9$ and $\tilde{l} = 1.35$ in a 280-rotor KL chain. Green dashed lines denote the predicted wave packet center position based on the group velocity at the excitation frequency in the infinite homogeneous chain and yellow solid lines indicate the fitted position of the kink center. The applied torque is over three times greater in (a) compared to (b).

Regarding strategies to optimize the coupling between the wave packet and the kink motion, the optimum scenario would correspond to the phonon wave packet transferring all of its energy to the zero-energy kink through the phonon-kink interaction. However, since kinks are subsonic solutions whose velocity cannot exceed the sound speed of the supporting system, there exists an upper bound on the energy that the KL chain kink can carry. Given this upper bound, the phonon wave packet can either be long compared to the kink, with lower energy density, or short, with higher energy density. In the case of a longer wave packet, the lower amplitude suggests weaker nonlinear effects, which are a necessary ingredient for the energy transfer. However, as energy density increases, the wave packet also moves away from the “phonon” limit considered herein.

Finally, we do not yet have a definitive answer to what configuration or phonon driving would yield the most efficient coupling. That said, as shown in the example in Fig. R31, higher driving amplitude can result in more efficient energy transfer. A systematic exploration of the optimal efficiency conditions is left for future work.

“(3) The experimental size of the KL chain seems highly limited by the need to Z-stack to avoid self-contact. Could the authors comments on the prospect of experimentally scaling these systems to useful dimensions?”

We thank the referee for this great question. To address this point, we propose a design that mitigates self-contact constraints, enabling longer experimental chains with fewer stacked layers. In addition, scaling of such rotary systems has been demonstrated in, *e.g.*, 3D printed gear-based mechanical metamaterials with gears down to 3.6 mm diameter [14].

A schematic illustrating this potential solution to reduce the z -stacking requirement is included in the SM.

SM:

III. PROSPECT OF EXPERIMENTALLY SCALING KL CHAINS TO USEFUL DIMENSIONS

Because the experimental size of our physical KL chain is strongly constrained by the need for z -stacking to avoid self-contact, we propose a potential solution to reduce the z -stacking requirement, as illustrated in Fig. R32. One could imagine the two fixed boundaries in grey are mounted to a substrate in the z - x plane. In addition, scaling of such rotary systems has been demonstrated in, *e.g.*, 3D printed gear-based mechanical metamaterials with gears down to 3.6 mm diameter [14].

FIG. R32. FIG. S57. Potential solution for reducing the z -stacking in realizing physical KL chains.

“(4) The authors make many comparisons between the KL chain and the discrete ϕ^4 dynamics. As the authors mention, another limit of the KL chain is sine-Gordon. Could the authors comments on why ϕ^4 , as opposed to sine-Gordon, is the right comparison to make?”

In this work, we investigated the F- and WF-phase kinks of the KL chain, while leaving the study of S-phase kinks for future work. Our focus on the F- and WF-phase kinks was motivated by the fact that, in the continuum limit, they are described by the ϕ^4 field theory. By contrast, the sine-Gordon (SG) model arises as the continuum limit of the S-phase kinks in the KL chain, where a full rotation of the rotors is required [1]. Importantly, there is no smooth transition between F-/WF-phase kinks and S-phase kinks, since they require different representations: for the F- and WF-phase kinks, all rotors are involved, whereas for the S-phase kinks only alternating sites (odd- or even-indexed rotors) participate. We anticipate exploring the S phase and the SG limit further in a future work.

“(5) Several aspects of Fig. 2 are hard for me to follow: Could the authors explain in more detail exactly what the “phonon spectrum” shown in Fig2 (c,f etc) is? How is it calculated? My interpretation is that first a nonlinear kink solution is numerically found using Newton Raphson. Next, the eigenmodes of the dynamics linearized about that kink state define this phonon spectrum. For me, the main text doesn’t explain this explicitly enough.”

We thank the reviewer for this comment and agree that the explanation in the original manuscript was not sufficiently clear. The referee’s interpretation is correct: We first obtain the kink solution numerically using the Newton–Raphson method, and then compute the spectrum of small oscillations around this equilibrium, which defines the phonon spectrum shown in Fig. 2. In the revised manuscript, we have clarified this point in the main text and added further details in the SM.

Main text: *“In this section, we numerically study the existence and discrete modal spectra of F- and WF-phase kinks for varying kink widths (keeping the lattice spacing a fixed). To vary the kink widths, we fix r and vary \bar{l} (equivalent to varying ψ in the homogeneous chain). The parameters are arbitrarily chosen as $a = 1$ m, $r = 1.5$ m, $k_e = 100$ N/m, and $m = 1$ kg, noting that all quantities will be normalized in subsequent analysis. For each parameter set, the corresponding kink state is obtained numerically using the Newton–Raphson method [12], and its discrete modal spectrum is then determined from the Jacobian matrix (see SM for additional details). As mentioned earlier, deep into the F phase ($d \ll 1$), the KL chain can be described by the continuum ϕ^4 model. To highlight the unique properties of the KL chain—specifically, the barrier-free propagation and stability of its zero-energy kink—in the SM, we compare them with those of the discrete ϕ^4 chain [11].”*

SM:

III. NEWTON–RAPHSON METHOD FOR STATIC KINK STATES, DISCRETE MODAL SPECTRA, AND MODE SHAPES

To compute the static kink configurations in the discrete ϕ^4 chain and the KL chain, as well as the associated discrete modal frequencies and mode shapes (except for kink center rotor angle sweep results in the KL chain), we proceed as follows. For each set of parameters, a static kink solution is obtained numerically using the Newton–Raphson method [12], starting from an appropriate initial guess.

For discrete ϕ^4 chains, the initial guess for an intersite-centered kink is taken from the anti-continuum limit [2] and represented by the sequence

$$(-u_0, \dots, -u_0, -u_0, u_0, u_0, \dots, u_0).$$

For onsite-centered kinks, we instead use

$$(-u_0, \dots, -u_0, -u_0, 0, u_0, u_0, \dots, u_0).$$

Analogous initial guesses are used in the KL chain. For intersite-centered kinks, we use

$$(\bar{\psi}, \dots, \bar{\psi}, \bar{\psi}, -\bar{\psi}, -\bar{\psi}, \dots, -\bar{\psi}),$$

and for onsite-centered kinks, we use

$$(\bar{\psi}, \dots, \bar{\psi}, \bar{\psi}, 0, -\bar{\psi}, -\bar{\psi}, \dots, -\bar{\psi}).$$

In cases where convergence is not achieved for the KL chain, we generate the initial guess using the transfer and inverse transfer functions of the chain (Eqs. S34, S35, S38, and S39), constrained to alternate rotor leaning directions as detailed in the previous section.

At each Newton-Raphson iteration, once the current state $\vec{\alpha} = [\alpha_1, \alpha_2, \dots, \alpha_N]$ is specified ($\alpha = u$ for the discrete ϕ^4 chain and $\alpha = \psi$ for the KL chain), the residual force/torque balance vector $R(\vec{\alpha})$ (force for the discrete ϕ^4 chain and torque for the KL chain) and the corresponding Jacobian $\mathbf{J}(\vec{\alpha}) = \partial R(\vec{\alpha})/\partial \vec{\alpha}$ are evaluated. The correction vector $\Delta \vec{\alpha} = [\Delta \alpha_1, \Delta \alpha_2, \dots, \Delta \alpha_N]$ is then obtained by solving

$$\mathbf{J}(\vec{\alpha}) \Delta \vec{\alpha} = -R(\vec{\alpha}), \quad \vec{\alpha} \leftarrow \vec{\alpha} + \Delta \vec{\alpha}, \quad (\text{R1})$$

and the iteration continues until $\|\Delta \vec{\alpha}\| \leq 10^{-8}$. Symmetry constraints are imposed near the kink center to stabilize the iteration. For example, for onsite-centered kinks, we enforce $\alpha_{n_{\text{kink}}} = 0$ and $\alpha_{n_{\text{kink}}-1} = -\alpha_{n_{\text{kink}}+1}$; for intersite-centered kink, we enforce $\alpha_{n_{\text{kink}}} = -\alpha_{n_{\text{kink}}+1}$, where n_{kink} denotes the kink center rotor index. The extent of the constraints (*i.e.*, how many rotors around the kink center are included) is adjusted depending on the quality of convergence.

Once a static kink configuration $\vec{\alpha}_{\text{kink}}$ is obtained, the equations of motion (Eqs. S27 and S14) are linearized about this equilibrium. This yields the Jacobian matrix \mathbf{J}_{kink}

$$\mathbf{J}_{\text{kink}} = \left. \frac{\partial R(\vec{\alpha})}{\partial \vec{\alpha}} \right|_{\vec{\alpha}=\vec{\alpha}_{\text{kink}}}, \quad (\text{R2})$$

which governs the evolution of small perturbations about $\vec{\alpha}_{\text{kink}}$. Solving the eigenvalue problem for \mathbf{J}_{kink} provides the discrete modal frequencies (eigenfrequencies), as well as the corresponding mode shapes (eigenvectors), of the kink state $\vec{\alpha}_{\text{kink}}$. The resulting eigenfrequency spectrum consists of extended phonon modes together with localized modes residing inside the band gap of infinite homogeneous chain (*i.e.*, in-gap modes).

“Fig. 2b and Fig. 2e are also hard to follow, and the caption of Fig. 2 jumps between figures somewhat out of order. I think Fig 2b is showing a 1D curve through 3D configuration space, coloured by both d and w_0 . But the aspect ratio of the 3D plot makes this hard to follow. For Fig. 2e, I am not entirely sure what is being shown, and the text is unclear. Perhaps a schematic figure would help here.”

The referee’s interpretation of the previous Fig. 2(b) is essentially correct: The 1D curve represents the scanned kink states in the 2D geometric phase space, with their kink width indicated by the z axis.

To improve clarity and better emphasize our main results in Fig. 2, we have moved this panel to the SM, along with the short-time dynamic simulations and the discrete ϕ^4 kink results. In addition, we have revised our previous

geometric phase diagram by providing an enlarged version of it, accompanied by a top-view projection of the same plot, which makes the transition lines and phase regions easier to follow. We have also modified the previous Fig. 2(e) (current Fig. R14(d)): instead of showing the kink center, we now plot the rotor angle at the kink center, which provides a clearer representation of the kink configuration. Furthermore, both the figure caption and the corresponding text in the main manuscript have been revised for improved clarity and conciseness. The revised content is as follows.

SM: “We introduce a new geometric phase diagram of the zero-energy kink in the KL chain parameterized by two dimensionless geometric variables, $\tilde{l} \equiv \bar{l}/a$ and $\tilde{r} \equiv r/a$, and overlay the widths of the F- and WF-phase kinks, as shown in Fig. S1.”

SM: “We note that Eq. S6 yields

$$d = \sqrt{1 + 4\tilde{r}^2 - \tilde{l}^2}, \quad (\text{R3})$$

which shows that multiple pairs of (\tilde{r}, \tilde{l}) can produce the same d , *i.e.*, this phase index alone is insufficient to determine the unit cell geometry in the KL chain, which consequently affects the kink’s dynamic properties (discussed further in a later section). In contrast, in Fig. S1, each point corresponds to a distinct unit cell geometry.”

FIG. R33. FIG. S1. Geometric phase diagram of the zero-energy kink, where the red line indicates $\tilde{r} = 1.5$, orange diamonds indicate kink states shown in Fig. 2(b,c), and the green star denotes the kink state examined in experiments. Left: Three-dimensional (3D) view. Right: Top view.

“More Minor comments:”

“(6) The units are hard to follow. For example, what does a kink velocity is -0.0885 (other examples across pages 6-7) mean? Is that big or small? I appreciate that these are simulation units, but some context for what this effectively dimensionless velocity means would be helpful.”

The kink velocity was obtained from the fitted slope of the traveled rotor index versus time. In the original manuscript, this quantity therefore has units of a/t (equivalently s^{-1} if the lattice constant a is used as the length unit), and a negative value indicates motion opposite to the direction of phonon propagation. To place this velocity in context, in the revised manuscript we normalize the kink velocity by the characteristic speed of sound in the system [1]. It is important to note that the “speed of sound” used for normalization here does not correspond to the group velocity at long wavelengths, which vanishes due to the presence of a finite cutoff frequency in the dispersion relation. Instead, it represents the asymptotic limit of the group velocity at short wavelengths in the near-continuum regime. In a discrete setting, even the maximum group velocity (typically occurring at midband frequencies) is smaller

than this limit. As a result, the normalized kink velocities mentioned in the revised main text in connection with the results shown in Fig. 4 are all below 0.1, which may appear small. However, this is consistent with the fact that kinks are subsonic solutions whose velocities range from zero up to the sound speed.

We have added the following sentence to the revised main text to clarify.

Main text: “In Fig. 4(a,b), we observe that as the driving torque increases ($\tilde{\tau}_0 = 0.0241$ and 0.0482), the kink propagates faster and radiates more in the direction opposite its motion. The kink velocity is extracted from the slope of the traveled rotor index versus time, normalized by the speed of sound (which represents the asymptotic limit of the group velocity at short wavelengths in the near-continuum regime [1]), yielding values of -0.0165 and -0.0581 , respectively. At the highest driving torque ($\tilde{\tau}_0 = 0.0718$), the kink motion becomes chaotic and is accompanied by significant energy radiation (Fig. 4(c)). As concerns the dependence of the interaction on frequency of the wave packet within the propagating band (middle row), with fixed torque amplitude ($\tilde{\tau}_0 = 0.0361$), we observed: differing wave packet dispersal (expected due to differing local band curvature and proximity to band edges), differing levels of kink radiation, and minimally modified (compared to wave packet amplitude effects) kink velocity. For the three cases tested, the normalized kink velocity ranged from -0.0316 (Fig. 4(d)) to -0.0078 (Fig. 4(f)). Finally, as concerns effects of the kink’s center position on the phonon-kink interaction dynamics (bottom row, $\tilde{\tau}_0 = 0.0361$), we observe different amounts of radiation from the kink (following wave packet interaction) and minimal changes in normalized kink velocity (-0.0441 to -0.0534). We note that while the kink velocities mentioned here may appear small, this is expected because, in the discrete setting, even the maximum group velocity remains below the characteristic sound speed. This is also consistent with the fact that kinks are subsonic solutions whose velocities range from zero up to the sound speed.”

IV. REPLY TO REFEREE 3

“The authors show, numerically and experimentally, that travelling wave packets (‘phonons’) can be used to control (move around, create) kinks in a Kane-Lubensky type mechanical metamaterial. This primary result is supported by extensive numerical simulations, investigating stability, phase diagrams, effect of damping, etc.”

“I think this is a very nice work and I recommend accepting with very minor suggestions (see later). It is very nice that a topological defect can be shuttled, in the lab, in a controllable way around a material, and I can see how this could lead to exciting applications (shaping proteins, computing, etc). It is also pretty nice that there are regimes of attractive and repulsive interactions, although it seems that the repulsive regime is a bit harder to achieve in practice.”

“The manuscript is well-written and the methodology is sound in both theory and experiments, adequately supporting the claims. I did not see any important detail missing to reproduce the work.”

We appreciate very much the positive comments from the referee. Below we respond to each question raised by the referee.

“Minor comments:”

“(a) In page 5 the authors mention ‘a small amount of phonons’. I’m pretty sure they do not have a small amount of phonons in such macroscopic, low-frequency experiment. I would expect the number of phonons to be on the order of 10^{25} . Perhaps the authors could drop or reduce the word ‘phonons’ there, or at least be clear that they are operating in a classical regime, and in no case should ‘small number of phonons’ be used in this setup.”

We thank the referee raising this issue, and agree with their point. To address this, we have clarified the usage of the term “phonon” in the abstract and at two locations in the main text, to distinguish our classical usage from its quantum-mechanical one. We have also replaced “phonons” with “energy” in the aforementioned sentence. In addition, we have reduced the use of “phonons” throughout the manuscript and have substituted it with “energy” or “waves” where appropriate. Furthermore, we have replaced the spatiotemporal response plots of normalized angular velocity with those of normalized total energy in the main text and in most parts of the Supplementary Materials (SM), to ensure consistency in presenting the results in terms of energy.

Abstract: “While phonon (or small-amplitude vibrational) wave packets are predicted to drive kink motion deterministically, experimental evidence has been elusive, with only stochastic motion from thermal phonons or quasi-static

loading observed.”

Main text: “Many studies have experimentally demonstrated that phonons (used herein in the classical sense as small-amplitude vibrational lattice waves) can initiate and affect the movement of mechanical kinks.”

Main text: “As noted, we refer to phonons here in the classical sense, as small amplitude vibrational wave packets, where small amplitude refers to small angular deformation (in particular compared to the kink).”

Main text: “...and the bottom row the spatiotemporal chain response in normalized total energy at each site (*i.e.*, the sum of kinetic and potential energy at each site normalized by the maximum in the full spatiotemporal field, denoted by \tilde{E}_n) during the wave packet generation, kink interaction, and subsequent scattering.”

Main text: “After the wave packet interacts with the kink, most of the energy is reflected (unlike the $d \rightarrow 0$, continuum limit, case), while a small amount of energy radiates from the kink, mostly opposite to the kink propagation.”

FIG. R34. **FIG. 3. Simulated phonon-kink interactions in KL chains with different unit cell geometries.** (Top row) Zoomed-in view of the onsite-centered kink state for KL chains with 280 rotors. (Middle row) Dispersion relation of infinite homogeneous chain with discrete modes of the finite chain overlapped at $k = \pi/a$ for each case in the top row. The black dashed line denotes the center frequency of the wave packet (middle of the phonon band). (Bottom row) Spatiotemporal response of normalized total energy at each site, with green dashed lines denoting the predicted wave packet center position based on the group velocity at the excitation frequency in the infinite homogeneous chain and yellow solid lines indicating the fitted center position of the kink (determined by fitting a hyperbolic tangent function to the chain configuration; see the SM for additional details). The first three columns from the left denote $\tilde{r} = 1.5$ at different d in the F/WF phase, and the rightmost $\tilde{r} = 0.8$ in the WF phase. $\textcircled{1}$ indicates the experimental onsite-centered kink state in Fig. 2.

★ $\tilde{r} = 1.5$, $d_{\text{exp}} = 2.0296$

FIG. R35. **FIG. 4. Simulated phonon-kink interaction in KL chains with the experimental unit cell geometry.** All panels show normalized total energy at each site (same color bar for all panels). (Top row) Kink dynamics under varying excitation amplitudes at a constant mid-band excitation frequency. **Insets shows zoomed-in results for the same time window.** (Middle row) Kink dynamics under different excitation frequencies (black dashed lines in the insets) at fixed excitation torque amplitude. (Bottom row) Kink dynamics at different kink center positions, under a constant mid-band excitation frequency and torque amplitude. Circled numbers indicate the kink states **with the same number in Fig. 2. The kink state is also shown in the insets.** $\Delta\tilde{\omega}$ represents the normalized phonon band width, defined as $|\tilde{\omega}_{k=0} - \tilde{\omega}_{k=\pi/a}|$. Green dashed lines denote **the predicted wave packet center position based on the group velocity at the excitation frequency in the infinite homogeneous chain** and yellow solid lines indicate the fitted center position of the kink.

“(b) [optional, would require larger edits] The paper does a significant amount of theory (stability analysis, phase diagrams, simulations, etc) before demonstrating the experimental control of kinks with phonons (Fig. 5) and the generation of kinks (Fig. 6). I find the stability analysis and so on good to support the key hypothesis. However, since phonon effects on boundary defects have been predicted elsewhere (including in some of the references), perhaps the manuscript would send a clearer message if the experiments were more prominent and the stability analysis and so on provided as extended data.”

In response to the referee’s suggestion, we made the following revisions. First, we performed additional experiments on kink control under different excitation amplitudes and revised the manuscript to make the experimental results more prominent (see Fig. R1). These new data further demonstrate that the excitation amplitude can serve as a tuning parameter for controlling kink propagation distance and velocity. Second, to better highlight the experimental contribution of this study, we added a photograph of the section of the experimental chain into Fig. 1(a) (Fig. R13[a]) in addition to the schematic that was present previously. Third, we streamlined Fig. 2 (Fig. R14) by moving the geometric phase diagram, the short-time dynamics, and the discrete ϕ^4 kink results to the SM. This reduces the emphasis on known theoretical background and allows us to better highlight our new results on internal mode families and their smooth variation with the kink center position before introducing the experimental findings. To our knowledge, neither the existence of internal mode families nor the smooth mode transitions as a function of the kink center position in the KL chain has been reported before. These transitions are likely connected to the rich phonon-kink interaction dynamics we observe (*e.g.*, attraction- and repulsion-type interactions, as well as distinct propagation behaviors). To further illustrate this connection, we have added an mode decomposition analysis of the time-domain results for both experiment and simulation in the SM (see Fig. R15).

-
- [1] B. G.-g. Chen, N. Upadhyaya, and V. Vitelli, *Proceedings of the National Academy of Sciences* **111**, 13004 (2014).
 - [2] M. Chirilus-Bruckner, C. Chong, J. Cuevas-Maraver, and P. Kevrekidis, *The sine-Gordon Model and its Applications: From Pendula and Josephson Junctions to Gravity and High-Energy Physics*, 31 (2014).
 - [3] C. R. Calladine, *International journal of solids and structures* **14**, 161 (1978).
 - [4] K. Sun, A. Souslov, X. Mao, and T. C. Lubensky, *Proceedings of the National Academy of Sciences* **109**, 12369 (2012).
 - [5] C. L. Kane and T. C. Lubensky, *Nature Physics* **10**, 39 (2014).
 - [6] X. Mao and T. C. Lubensky, *Annual Review of Condensed Matter Physics* **9**, 413 (2018).
 - [7] Y. S. Kivshar and D. K. Campbell, *Physical Review E* **48**, 3077 (1993).
 - [8] O. M. Braun and Y. S. Kivshar, *The Frenkel-Kontorova model: concepts, methods, and applications*, Vol. 18 (Springer, 2004).
 - [9] P. G. Kevrekidis and J. Cuevas-Maraver, *Past, Present and Future* **26** (2019).
 - [10] P. Kevrekidis and M. Weinstein, *Physica D: Nonlinear Phenomena* **142**, 113 (2000).
 - [11] T. Dauxois and M. Peyrard, *Physics of solitons* (Cambridge University Press, 2006).
 - [12] A. Gilat and V. Subramaniam, *An Introduction with Applications Using MATLAB*, **20014** (2013).
 - [13] Y. Kamimura, K. Edagawa, and S. Takeuchi, *Acta materialia* **61**, 294 (2013).
 - [14] X. Fang, J. Wen, L. Cheng, D. Yu, H. Zhang, and P. Gumbsch, *Nature Materials* **21**, 869 (2022).

Response to Referees. Referee comments are in blue text. Replies to referee comments are in black text. Revised text incorporated into the updated manuscript is in red text.

I. REPLY TO REFEREE 1

“I have reviewed the response and changes made by the authors. They have significantly improved their manuscript, which I now recommend for publication.”

We thank the reviewer for recognizing the improvements in our revised manuscript and for recommending it for publication.

II. REPLY TO REFEREE 2

“The authors have provided a substantial amount of work to address my comments. I repeat my recommendation for publication in Nature Communications.”

We appreciate the reviewer’s reevaluation and their continued recommendation for publication.

III. REPLY TO REFEREE 3

“The authors have addressed (most) of my concerns, and I think most of the concerns from the other reviewers. I’m particularly impressed that they manage to experimentally demonstrate most of the soliton phenomena that they had predicted theoretically—even though I would not have expected it was necessary for publication.”

We thank the referee for the effort and time they put in reviewing our revised manuscript.

“I still struggle a bit with the fact that the authors now redefine the word phonon.”

“For 100 years, the word phonon has referred to a unit of energy in a specific mode (ie. $\hbar\omega$, which for the author’s system would be c.a. 10^{-30} J). Phonons do not look like wavepackets, they look like standing waves — although you can combine them to form a wavepacket, but then they are longer a pure phonon. A person with a traditional physics training, when said that ‘phonons’ are put into the system, will think that the system is oscillating with a standing wave with amplitudes around 10^{-20} meters.”

“If the authors want to say that they put wavepackets, why don’t call them wavepackets? If long-term, quantum applications of this idea are on the horizon (which is fine), it is OK to add a paragraph in the outlook that maybe one day this could be made with single phonons and this is an area to be studied.”

We understand the referee’s concern regarding the usage of the term phonon, and agree with their definition as a quanta of vibrational energy. However, we disagree with the definition of a phonon as a standing wave, as, even at the quantum level, a single phonon would represent a traveling wave, whereas a standing wave would be a superposition of two of these counter-propagating waves. Further, a wave packet can be assembled via the superposition of even a relatively few phonon modes with appropriate phase delays.

Our intent was not to redefine the term “phonon”, rather, to use it in the sense that has been widely adopted in the classical acoustics and mechanics communities to describe acoustic or elastic wave excitations in structured media, e.g., Ref. [1].

In any case, we have revised our manuscript to clarify our contribution as “observation of mechanical kink control and generation via acoustic wave packets.” We have adjusted this wording throughout the manuscript, included sentences drawing a connection between computational studies of “phonon-kink” interaction, and included a sentence in the conclusion that suggests their future possible extension to the quantum scales. These revisions are as follows.

Title: “Observation of mechanical kink control and generation via **acoustic waves**”

Abstract: “While **acoustic wave packets** (here defined as **small-amplitude mechanical waves, sometimes referred to as phonons**) have been predicted to drive kink motion deterministically, experimental evidence has been elusive, with only stochastic motion from thermal phonons or quasi-static loading observed.”

Main Text:

“Many studies have experimentally demonstrated that **acoustic wave packets** (here defined as **small-amplitude, near-linear mechanical waves, sometimes referred to as phonons in their classical sense [1]**) can initiate and affect the movement of mechanical kinks.”

“Theoretical and computational studies have given further insight to this question, exploring the interaction between **acoustic waves** and mechanical kinks via canonical models such as the ϕ^4 [2–9] and sine-Gordon (sG) systems [10].”

FIG. R1. **Overview of a small-amplitude acoustic wave packet moving the static zero-energy kink in the KL chain.** (a) Schematic of a chain section, where black lines represent massless rotors with radius r and lattice constant a , blue circles represent point masses m , and red lines represent linear normal springs with spring constant k_e and instantaneous length $l_{n,n+1}$ (with n and $n + 1$ as rotor indices), accompanied by a photograph of the corresponding experimental chain section. (b) Kink state before (top) and after (bottom) **acoustic-wave-kink** interaction. The yellow dashed circle indicates the center of the kink.

“Finally, we suggest our findings may lead to analogs in quantum regimes for controlling kinks with negligible PN barriers via “true” phonons, including the incorporation of phenomena unique to such scales (*e.g.*, tunnel nucleation of kinks [11]).”

-
- [1] M. I. Hussein, M. J. Leamy, and M. Ruzzene, *Applied Mechanics Reviews* **66**, 040802 (2014).
 - [2] W. Hasenfratz and R. Klein, *Physica A: Statistical Mechanics and its Applications* **89**, 191 (1977).
 - [3] Y. Wada and J. Schrieffer, *Physical Review B* **18**, 3897 (1978).
 - [4] N. Theodorakopoulos, *Zeitschrift für Physik B Condensed Matter* **33**, 385 (1979).
 - [5] N. Theodorakopoulos, W. Wunderlich, and R. Klein, *Solid State Communications* **33**, 213 (1980).
 - [6] R. Klein, W. Hasenfratz, N. Theodorakopoulos, and W. Wunderlich, *Ferroelectrics* **26**, 721 (1980).
 - [7] H. Ishiuchi and Y. Wada, *Progress of Theoretical Physics Supplement* **69**, 242 (1980).
 - [8] M. Ogata and Y. Wada, *Journal of the Physical Society of Japan* **53**, 3855 (1984).
 - [9] A. Abdelhady and H. Weigel, *International Journal of Modern Physics A* **26**, 3625 (2011).
 - [10] N. Theodorakopoulos, in *Ordering in Strongly Fluctuating Condensed Matter Systems* (Springer, 1980) pp. 145–149.
 - [11] B. Petukhov, *Journal of Statistical Mechanics: Theory and Experiment* **2018**, 093104 (2018).